# Glucose transporter 10 modulates adipogenesis via an ascorbic acid-mediated pathway to protect mice against diet-induced metabolic dysregulation

**Chung-Lin Jiang[1], Wei-Ping Jen[1], Chang-Yu Tsao[1], Li-Ching Chang[2], Chien-Hsiun Chen[2], Yi-Ching Lee[1] ***

**1** Institute of Cellular and Organismic Biology, Academia Sinica, Taipei, Taiwan, **2** Institute of Biomedical Sciences, Academia Sinica, Taipei, Taiwan

* yiching@gate.sinica.edu.tw

**Data Availability Statement:** All relevant data are within the manuscript and its Supporting Information files.

## Abstract

The development of type 2 diabetes mellitus (T2DM) depends on interactions between genetic and environmental factors, and a better understanding of gene-diet interactions in T2DM will be useful for disease prediction and prevention. Ascorbic acid has been proposed to reduce the risk of T2DM. However, the links between ascorbic acid and metabolic consequences are not fully understood. Here, we report that glucose transporter 10 (GLUT10) maintains intracellular levels of ascorbic acid to promote adipogenesis, white adipose tissue (WAT) development and protect mice from high-fat diet (HFD)-induced metabolic dysregulation. We found genetic polymorphisms in *SLC2A10* locus are suggestively associated with a T2DM intermediate phenotype in non-diabetic Han Taiwanese. Additionally, mice carrying an orthologous human *Glut10*$^{G128E}$ variant (*Glut10*$^{G128E}$ mice) with compromised GLUT10 function have reduced adipogenesis, reduced WAT development and increased susceptibility to HFD-induced metabolic dysregulation. We further demonstrate that GLUT10 is highly expressed in preadipocytes, where it regulates intracellular ascorbic acid levels and adipogenesis. In this context, GLUT10 increases ascorbic acid-dependent DNA demethylation and the expression of key adipogenic genes, *Cebpa* and *Pparg*. Together, our data show GLUT10 regulates adipogenesis via ascorbic acid-dependent DNA demethylation to benefit proper WAT development and protect mice against HFD-induced metabolic dysregulation. Our findings suggest that *SLC2A10* may be an important HFD-associated susceptibility locus for T2DM.

## Author summary

Environmental triggers may amplify genetically determined disease susceptibility, especially for carriers of rare variants with relatively large individual effect sizes, making these polymorphisms highly informative for predicting individualized clinical risk and preventing disease. Since transitions in dietary pattern have greatly contributed to the increased

**Funding:** This study was supported by grants from Academia Sinica, Taiwan (AS 105-TP-B04) and the Ministry of Science and Technology (MOST), Taiwan (MOST 107-2320-B-001-024 and MOST 108-2320-B-001-022) to YCL. The funders had no role in study design, data collection and analysis, decision to publish, or preparation of the manuscript.

**Competing interests:** The authors have declared that no competing interests exist.

prevalence of obesity and accelerated the spread of the T2DM epidemic worldwide, a better understanding of gene-diet interactions in T2DM will be useful for disease prediction and prevention. Here, we demonstrate that polymorphisms in the gene encoding GLUT10 are associated with a T2DM intermediate phenotype in non-diabetic human subjects. Additionally, mice that carry a GLUT10 rare variant have reduced WAT development and are susceptible for HFD-induced T2DM. We further demonstrate that GLUT10 is highly expressed in preadipocytes, where it regulates intracellular ascorbic acid levels and ascorbic acid-dependent DNA demethylation to control adipogenesis. Preadipocytes carrying the GLUT10 rare variant or with knockdown of GLUT10 expression have reduced the adipogenesis. Thus, we are able to conclude that GLUT10 regulates adipogenesis via ascorbic acid-dependent DNA demethylation to affect WAT development and contribute to the sensitivity of HFD-induced metabolic dysregulation.

## Introduction

Large-scale genome-wide association studies (GWAS) have successfully identified many genes that contribute to type 2 diabetes mellitus (T2DM); however, those genes explain only a small proportion of the disease susceptibility and heritability [1, 2]. Since transitions in dietary pattern have greatly contributed to the increased prevalence of obesity and accelerated the spread of T2DM epidemic worldwide, it is clear that environmental factors heavily affect the risk of T2DM. Therefore, the gene-diet interactions may be crucial to accelerate development of T2DM. Additionally, rare variants in specific genes, which cannot easily be identified via GWAS, might have strong effects on T2DM risk [3–5].

Ascorbic acid has been implied to reduce risk of T2DM [6]. Epidemiological studies have shown that plasma ascorbic acid levels are inversely correlated with waist-to-hip ratio [6–9], which reflects the accumulation of visceral fat and is highly associated with the risk of T2DM [10]. Furthermore, animal studies demonstrated that ascorbic acid supplementation reduces inflammation and adiposity in high-fat-diet (HFD)-fed animals [6, 11]. However, clinical trials testing ascorbic acid supplementation for the prevention or treatment of obesity and its co-morbidities have produced inconsistent results [6, 12]. Thus, the possible mechanisms and physiological effects that link ascorbic acid with reduced risk of T2DM require further exploration.

Ascorbic acid is an antioxidant, and it is also an enzyme co-factor contributing in a broad range of biological processes [13]. Ascorbic acid and its oxidized form, dehydroascorbic acid (DHA), can be transported into cells and intracellular compartments by active sodium ascorbic acid transporters (SVCTs) and facilitative glucose transporter members (GLUTs), respectively [14]. Loss-of-function mutations in GLUT10 lead to a rare autosomal-recessive connective tissue disorder called arterial tortuosity syndrome (ATS; OMIM 208050) [15]. We and others have demonstrated that GLUT10 is a critical factor in the maintenance of intracellular ascorbic acid levels and redox hemostasis in vitro and in vivo [16–20]. GLUT10 is localized to the endomembrane system, mitochondria and the nuclear envelope, and it mediates intracellular DHA transport in aortic smooth muscle cells (ASMCs), adipocytes and fibroblasts [14–16, 19]. The DHA transported into subcellular compartments can be regenerated into ascorbic acid, which enhances cellular uptake of DHA as a mechanism to maintain intracellular ascorbic acid levels and redox homeostasis [16–21]. Therefore, GLUT10 may play a critical role in ascorbic acid homeostasis under stress conditions [16, 20]. Ascorbic acid is essential for hydroxylation of prolyl and lysyl residues during collagen synthesis and assembly. The effects

of GLUT10 on ascorbic acid-dependent collagen synthesis/assembly and redox balance might explain connective tissues abnormalities observed in mice with functional GLUT10 deficiency and ATS patients [15, 17, 20].

Interestingly, the chromosomal region around GLUT10 the gene (*SLC2A10)* has been associated with T2DM in French and Finnish sib-pairs though linkage analysis [22, 23]. However, a direct association between the genetic polymorphisms in *SLC2A10* locus and T2DM has not been found with GWAS [24–29]. The reasons for this discrepancy are unknown but may be related to the physiological role of GLUT10. Notably, high expression of GLUT10 was detected in white adipose tissue (WAT), especially in stromal vascular fraction (SVF) cells in both humans and mice [16, 30]. Adipose tissue serves as a major energy reserve, and it is an active endocrine organ secreting various adipokines to regulate whole-body energy homeostasis. Thus, either an excess or a deficiency of adipose tissue may have harmful metabolic consequences [31]. Based on these known GLUT10 functions, the expression pattern, and its potential association with T2DM, we suspected that GLUT10 might have roles in metabolism and contribute to T2DM.

Here, we used population studies, a mouse model carrying a GLUT10 rare variant (*Glut10^{G128E}* mice) and *in vitro* adipogenesis experiments to test this idea. We demonstrate that genetic variants in *SLC2A10* locus are associated with T2DM-related intermediate traits in human subjects. Furthermore, *Glut10^{G128E}* mice have reduced adipogenesis and WAT development and are susceptible to HFD-induced metabolic dysregulation. Finally, we demonstrate GLUT10 modulates ascorbic acid-mediated DNA-demethylation and gene expression of *Cebpa* and *Pparg* to affect adipogenesis. Together our results demonstrate that GLUT10 positively regulates adipogenesis, WAT development and has beneficial effects on metabolism. Our findings also suggest that *SLC2A10* may be an important susceptibility locus for HFD-induced T2DM.

## Results

### *SLC2A10* is suggestively associated with T2DM intermediary traits

We first examined whether genetic polymorphisms of *SLC2A10* might be associated with T2DM intermediate traits that reflect an increased risk for the disease, as a direct association between *SLC2A10* genetic polymorphisms and T2DM has not been found with GWAS. By analyzing the GWAS Central database (https://www.gwascentral.org) [32], we found that several genetic polymorphisms in *SLC2A10* locus are modestly associated with multiple T2DM-related phenotypes, including fasting plasma glucose, fasting insulin and body mass index in non-diabetic populations of independent human populations and studies (S1 Table). To further validate these results, we examined the association of *SLC2A10* genetic polymorphisms with glycated hemoglobin level (HbA1c), a well-known risk factor for developing T2DM [33] in non-diabetic population of Taiwan Super Control Study [34, 35]. To increase the power and better detect genetic effects, we included the imputed single nucleotide polymorphisms (SNPs) [34]. Using this approach, we were able to identify genetic polymorphisms in *SLC2A10* were suggestively associated with HbA1c level (the SNP with the best association had *P*-value = $1.11 \times 10^{-5}$) (Table 1).

### *Glut10^{G128E}* mice are susceptible to HFD-induced glucose intolerance and insulin resistance

To directly test whether GLUT10 may play a role in T2DM, we examined whether metabolism might be affected in mice carry a GLUT10 variant with known impacts on cellular physiology.

**Table 1. HbA1c-associated variants in the *SLC2A10* locus among Han Taiwanese.**

| rsID | Other Allele A | Effect Allele B | EAF | AA | AB | BB | BETA | SE | *P*-value |
|---|---|---|---|---|---|---|---|---|---|
| rs116953726 | A | T | 0.00158 | 890.18 | 2.82 | 0.00 | -1.3089 | 0.38405 | 1.11E-05 |
| rs189215420 | C | T | 0.00437 | 885.19 | 7.81 | 0.00 | -0.82075 | 0.17439 | 1.92E-05 |
| rs150591187 | G | A | 0.00214 | 889.17 | 3.83 | 0.00 | -1.4658 | 0.27307 | 5.30E-05 |
| rs185800001 | T | C | 0.00222 | 889.04 | 3.96 | 0.00 | -1.0507 | 0.22167 | 9.65E-05 |
| rs180994914 | A | C | 0.00158 | 890.18 | 2.82 | 0.00 | -1.4413 | 0.41693 | 0.0016818 |
| rs182118368 | C | T | 0.00181 | 889.76 | 3.24 | 0.00 | -0.91829 | 0.27623 | 0.0017609 |
| rs181263431 | A | G | 0.00167 | 890.01 | 2.99 | 0.00 | -1.4261 | 0.42028 | 0.0027062 |
| rs148807321 | T | A | 0.00197 | 889.49 | 3.51 | 0.00 | -0.64823 | 0.38608 | 0.012209 |
| rs139228424 | G | A | 0.01534 | 865.69 | 27.20 | 0.10 | -0.26393 | 0.11854 | 0.014201 |
| chr20:45304200:I | C | CT | 0.00830 | 878.48 | 14.22 | 0.30 | -0.31174 | 0.14525 | 0.016956 |
| rs117446254 | G | C | 0.02608 | 847.57 | 44.26 | 1.16 | -0.1893 | 0.078804 | 0.023363 |
| chr20:45373131:D | AGAG | A | 0.00696 | 880.58 | 12.40 | 0.02 | -0.32957 | 0.14587 | 0.028406 |
| rs13043514 | G | A | 0.06696 | 779.58 | 107.16 | 6.21 | -0.10268 | 0.050914 | 0.035145 |
| rs760874 | G | A | 0.36628 | 352.90 | 426.01 | 114.08 | -0.038666 | 0.018619 | 0.038189 |
| rs141885539 | A | G | 0.00307 | 887.52 | 5.48 | 0.00 | -0.61586 | 0.23967 | 0.042469 |
| rs6124865 | T | C | 0.37002 | 348.69 | 427.74 | 116.55 | -0.037797 | 0.018642 | 0.042896 |

EAF = frequency of the effect allele

BETA = effect on HbA1c for each copy of the effect allele

SE = standard error of effect size

The *Glut10*[G128E] mouse strain was generated by N-ethyl-N-nitrosourea (ENU) mutagenesis [36], and the same human rare variant (rs1226198239) was identified and predicted to be benign (S2 Table, heighted in yellow). The variant encodes a protein with an altered residue in extracellular loop 5 that is conserved in mammals (S1A and S1B Fig). GLUT10[G128E] protein has impaired mitochondrial targeting, decreased DHA uptake, and decreased intracellular ascorbic acid levels in ASMCs [16, 20]. We backcrossed the *Glut10*[G128E] mice from their original C3H background onto the C57BL/6J background that is widely used in metabolic disease research. *Glut10*[G128E] mice on a C57BL/6J had similar phenotypes to those previously observed on the C3H background. *Glut10*[G128E] mice have a normal appearance and do not exhibit signs or pathological features of ATS-related major artery phenotypes at birth. The mice develop observable pathology in vascular wall of major arteries at 10 months of age [20, 36]. We therefore used the *Glut10*[G128E] mice as a model to study how an *SLC2A10* polymorphism might affect metabolism in mice.

We compared metabolism-related parameters in *Glut10*[G128E] and WT mice on a normal chow diet with *ad libitum* feeding (CD) at 20 weeks of age. *Glut10*[G128E] mice had no significant differences from WT in body weight gain or fasting glucose levels (Fig 1A, 1B and 1C). Interestingly, *Glut10*[G128E] mice did exhibit significantly higher HbA1c levels (Fig 1D), similar to the findings in our human population study. Despite this increased level of HbA1c, *Glut10*[G128E] mice maintained normal fasting insulin levels (Fig 1E) and showed no difference from WT in their ability to clear a glucose bolus or insulin sensitivity, which were respectively measured by the glucose tolerance test (GTT) and insulin resistance test (ITT) (Fig 1F and 1G). Aging is a major risk factor for metabolic dysregulation, so we further examined mice on CD at 24 months of age. *Glut10*[G128E] mice exhibited fasting blood glucose levels and GTT results similar to WT mice (Fig 1H). Collectively, our data showed that while *Glut10*[G128E] mice have higher HbA1c levels, no obvious metabolic abnormalities are present when the mice are fed with CD.

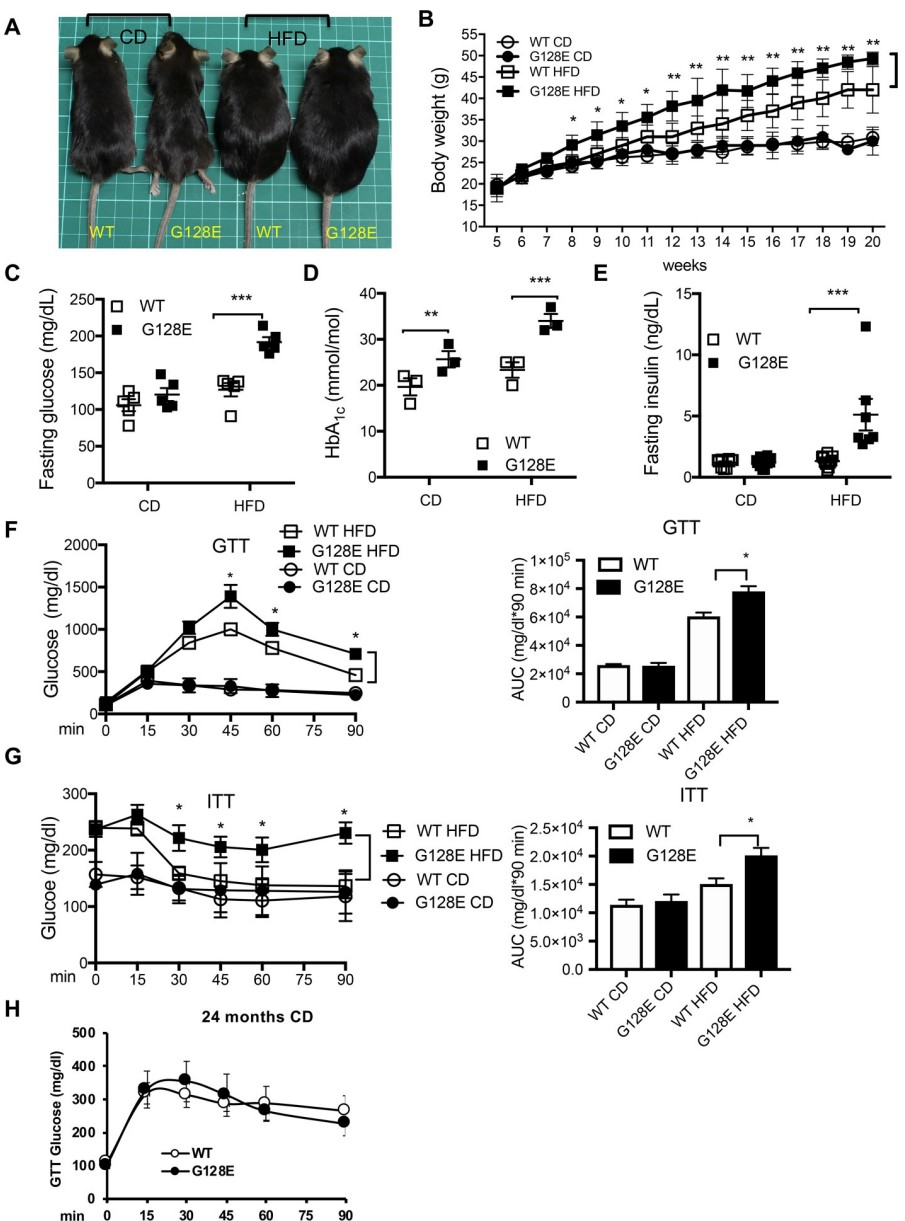

**Fig 1. *Glut10^{G128E}* mice are predisposed to HFD-induced glucose intolerance and insulin resistance.** *Glut10^{G128E}* mice and WT mice were fed with a CD or HFD from 5 to 20 weeks of age. Data were collected from the mice at indicated of age or at the conclusion of feeding (20 weeks of age). (A) Representative photograph of mice at 20 weeks of age. Each square on the green mat is 1 cm × 1 cm. (B) Body weights are shown for indicated ages. n = 20–40 mice per group. (C) Fasting glucose, (D) HbA1c levels, and (E) fasting insulin levels were measured at 20 weeks of age. (F) Glucose tolerance test (GTT) was performed in 16-week-old mice and (G) insulin tolerance test (ITT) was performed in 18-week-old animals. Right panels in F and G respectively show the areas under the GTT and ITT curves (AUC). n = 6 mice per group. (H) GTT was performed in WT and *Glut10^{G128E}* mice on a CD at 24 months of age. n = 6 mice per group. The data are shown as mean ± SEM. $^*P < 0.05$, $^{**}P < 0.01$, $^{***}P < 0.001$.

We then examined the metabolic consequences of feeding *Glut10^{G128E}* mice and WT with a HFD from 5 weeks of age to 20 weeks of age. Surprisingly, on a HFD, *Glut10^{G128E}* mice gained more weight than WT mice (Fig 1B), despite both genotypes exhibiting comparable food intake, physical activity, energy expenditure, and respiratory exchange ratio (RER) on either a

CD or HFD (S2 Fig). However, trends toward reductions in physical activity, VO2, VCO2 and heat production were observed in *Glut10*$^{G128E}$ mice on a HFD (S2B–S2F Fig). On the HFD, *Glut10*$^{G128E}$ mice had significantly higher fasting blood glucose, HbA1c levels, and fasting insulin levels compared to WT mice (Fig 1C, 1D and 1E). Furthermore, *Glut10*$^{G128E}$ mice exhibited worse glucose tolerance and insulin sensitivity than WT mice (Fig 1F and 1G). We then examined insulin sensitivity in insulin-responsive organs in HFD-fed WT and *Glut10*$^{G128E}$ mice by probing insulin-induced AKT phosphorylation, a downstream target of the insulin receptor. After insulin injection, phosphorylated AKT was highly elevated in epididymal WAT (eWAT), liver, and skeletal muscle of HFD-fed WT mice (S3A–S3C Fig). In contrast, the insulin induction of AKT phosphorylation was attenuated in eWAT of *Glut10*$^{G128E}$ mice on HFD (S3A Fig). Of note, AKT phosphorylation in eWAT of *Glut10*$^{G128E}$ mice on CD was also lower than WT controls (S3A Fig). In summary, our data showed that *Glut10*$^{G128E}$ mice are predisposed to HFD-induced weight gain, glucose intolerance, and insulin resistance.

### *Glut10*$^{G128E}$ mice are susceptible to HFD-induced inflammation and fibrosis in eWAT, adipokine dysregulation, and ectopic lipid deposition

We next asked how the *Glut10*$^{G128E}$ variant might predispose mice to HFD-induced metabolism dysregulation. We compared the gross morphologies of T2DM-related tissues and organs in *Glut10*$^{G128E}$ and WT mice on a CD. Notably, *Glut10*$^{G128E}$ mice showed reduced size and weight of visceral epididymal WAT (eWAT) on CD (Fig 2A and 2B). In contrast, the morphology, mass, and histology of subcutaneous inguinal WAT (sWAT), interscapular brown adipose tissue (iBAT), liver, and pancreas of *Glut10*$^{G128E}$ mice all appeared normal on CD (Fig 2A and 2B; S4A–S4D Fig). Histology analysis of eWATs revealed a reduced average adipocyte cell size in *Glut10*$^{G128E}$ eWATs that was mainly attributable to an increase in the proportion of small adipocytes (Fig 2C, 2D and 2E). Accordingly, *Glut10*$^{G128E}$ mice exhibited low levels of lipid storage in eWAT (Fig 2F) and the body fat composition (Fig 2G), while body lean composition was not different from WT controls (Fig 2H). Thus, the most apparent phenotypes in CD-fed *Glut10*$^{G128E}$ mice were observed in eWAT.

The eWAT in *Glut10*$^{G128E}$ mice was then compared with that in WT mice on a HFD. *Glut10*$^{G128E}$ mice had slightly lower mass of eWAT (Fig 2A and 2B) but no significant difference from controls in lipid storage on a HFD (Fig 2F). Histological analyses showed that eWAT from HFD-fed *Glut10*$^{G128E}$ mice had high heterogeneity in its cell size distribution and no significant difference from WT in average adipocyte size (Fig 2C, 2D and 2E). The appearance crown-like structures were frequently observed in on HFD-fed *Glut10*$^{G128E}$ eWAT (Fig 2C). The structures indicating macrophage accumulation is a hallmark of adipose tissue inflammation and contributes to tissue fibrosis. Macrophage infiltration and fibrosis were confirmed by CD68 staining, a protein highly expressed in macrophages, and Masson's trichrome staining for collagen. The results revealed increased macrophage infiltration and extracellular matrix deposition in *Glut10*$^{G128E}$ eWAT (Fig 2C). Moreover, mRNA expression of macrophage marker, EGF-like module-containing mucin-like hormone receptor-like 1 (EMR1; also known as F4/80), and fibrosis-promoting factor, transforming growth factor β (TGF-β), were markedly increased in eWAT of HFD-fed *Glut10*$^{G128E}$ mice (S5A Fig). We conclude that *Glut10*$^{G128E}$ mice are susceptible to HFD-induced eWAT inflammation and fibrosis.

WAT secretes biologically active adipokines, such as adiponectin, leptin, tumor necrosis factor-α (TNF-α), and interleukin-6 (IL-6), which control systemic energy homeostasis and are often affected by pathophysiological conditions [37]. The serum level of adiponectin was reduced, while serum levels of leptin, IL-6 and TNF-α were higher in HFD-fed *Glut10*$^{G128E}$ mice (Fig 2I–2L). Interestingly, the serum levels of adiponectin and IL-6 were also altered in

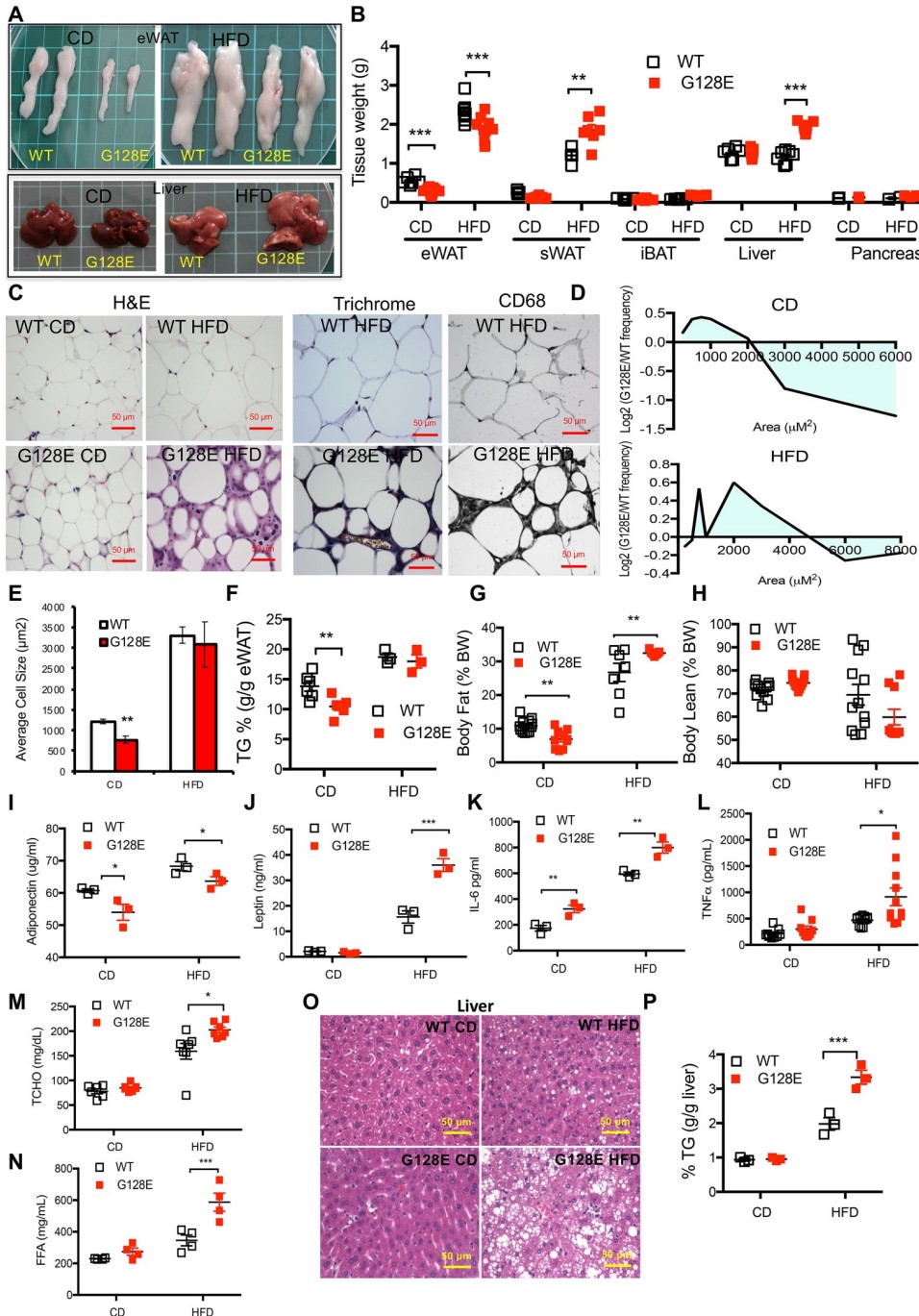

**Fig 2. *Glut10^{G128E}* mice are susceptible to HFD-induced inflammation and fibrosis in eWAT as well as ectopic lipid deposition.** *Glut10^{G128E}* mice and WT mice were fed a CD or HFD from 5 to 20 weeks of age. Data were collected from mice at 20 weeks of age. (A) Representative photographs of eWAT and liver from experimental mice. Each square on the green mat is 1 cm × 1cm. (B) Tissue weights. (C) Hematoxylin and eosin (H&E) staining, trichrome staining, and CD68 immunohistochemistry of eWAT sections. (D) The cross-sectional areas of adipose cells in eWAT are presented as the $\log_2$ ratio of cells in a particular size range in *Glut10^{G128E}* mice relative to that in WT mice on a CD or HFD. (E) The average adipocyte size in eWAT. D and E, more than 1000 adipocytes were measured from the eWAT sample from each mouse. n = 6 mice per group. (F) Triglyceride (TG) content in eWAT. (G and H) Body fat and body lean compositions. (I-L) The adiponectin, leptin, IL-6, and TNFα levels in serum. (M) Total cholesterol (TCHO) levels and (N) free fatty acid (FFA) levels in serum. (O) H&E staining of liver sections. (P) Triglyceride (TG) content in liver. Data are shown as the mean ± SEM. $^*P < 0.05$, $^{**}P < 0.01$, $^{***}P < 0.001$.

                                          

CD-fed $Glut10^{G128E}$ mice (Fig 2I and 2K). The mRNA expression levels for these adipokines in $Glut10^{G128E}$ eWAT were consistent with the serum levels on both CD and HFD (S5A and S5B Fig). Thus, $Glut10^{G128E}$ mice have altered expression of adipokines on CD, and the adipokine dysregulation is further augmented on HFD.

Inflammation and fibrosis in eWAT and adipokine dysregulation can contribute to ectopic lipid deposition in other tissues [38–40]. As such, we compared ectopic lipid deposition in $Glut10^{G128E}$ mice and WT mice on a HFD. The serum levels of total cholesterol and free fatty acids were significantly increased in $Glut10^{G128E}$ mice (Fig 2M and 2N). Ectopic lipid accumulation was observed in the liver of HFD-fed $Glut10^{G128E}$ mice, as demonstrated by the increased size, weight and triglyceride content of the liver, along with an increase in liver fat vacuoles (Fig 2A, 2B, 2O and 2P). Ectopic lipid accumulation was also observed in the iBAT of $Glut10^{G128E}$ mice, as increased occurrence of fat vacuoles was observed (S4B Fig). The size and mass of sWAT were also increased in $Glut10^{G128E}$ mice (S4A Fig and Fig 2B). However, no obvious histological changes were observed in sWAT of $Glut10^{G128E}$ mice (S4C Fig). Overall, hyperlipidemia and ectopic lipid deposition were presented in HFD-fed $Glut10^{G128E}$ mice.

The pancreas appeared to be unaffected in $Glut10^{G128E}$ mice on either a CD or HFD, according to insulin secretion, tissue morphology and mass, and histology of pancreatic islets (Fig 1E, Fig 2B and S4A and S4D Fig). Furthermore, no obvious vascular abnormalities were observed in major arteries or adipose tissues of $Glut10^{G128E}$ mice on a CD or HFD, based on histology and functional architecture of the adipose vasculature revealed by fluorescent microbead perfusion (S6 Fig).

Together, $Glut10^{G128E}$ mice exhibit reduced eWAT and irregular adipokine expression on CD; the mice are further predisposed to HFD-induced eWAT inflammation and fibrosis, augmented adipokine dysregulation, hyperlipidemia and ectopic lipid accumulation.

## $Glut10^{G128}$ mice have reduced eWAT and altered adipokine profiles at an early age

Because $Glut10^{G128E}$ mice exhibit significantly reduced mass and cell size of eWAT coincident with irregular adipokine levels on CD, which are critical pathophysiological mechanism underlying HFD-induced metabolic dysregulation. We therefore compared eWAT and adipokine expression in $Glut10^{G128E}$ mice and WT mice at an early age. In mice, WAT compartments begin to develop during the late gestational stages and then exhibit a marked postnatal expansion just before 10 weeks of age [41]. GLUT10 expression was detected at high levels in eWAT of juvenile mice (3 weeks) compared with adult mice on CD (20 weeks) (Fig 3A). Interestingly, $Glut10^{G128E}$ mice exhibited compromised eWAT development at 3 weeks of age, as indicated by reduced mass, reduced triglyceride content, and reduced average adipocyte size with an increase small adipocyte population (Fig 3B–3F). Consistent with reduced eWAT in $Glut10^{G128E}$ mice, serum levels of most adipokines were reduced, while IL-6 levels were increased, according to the results of an adipokine array (Fig 3G and 3H). The changes in adiponectin and IL-6 expression were validated in serum by ELISA and mRNA expression in eWAT by RT-PCR (Fig 3I and 3J). These results demonstrated that $Glut10^{G128E}$ mice have reduced eWAT and altered adipokine profile at an age as early as 3 weeks.

## GLUT10 is required for adipocyte differentiation *in vivo*

We next explored the potential mechanism through which GLUT10 regulates eWAT development. We performed RNA sequencing (RNA-seq) analysis of eWAT from 3-week-old WT and $Glut10^{G128E}$ mice to profile the genes affected by the GLUT10 variant. Ingenuity pathway analysis (IPA) revealed that the GLUT10 variant affected genes involved in adipogenesis,

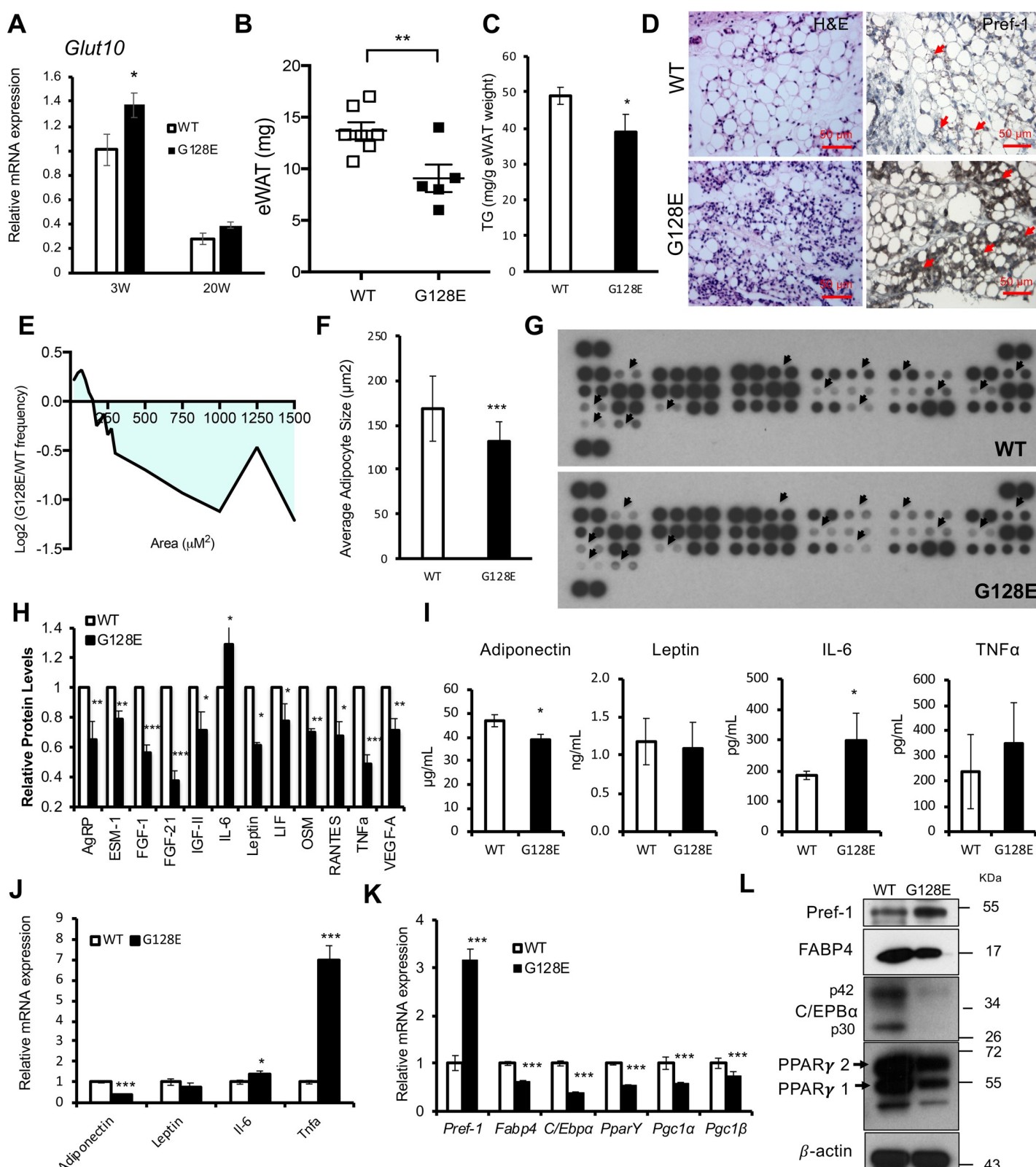

**Fig 3. *Glut10^{G128}* mice have compromised eWAT development and altered adipokine profiles at an early age.** (A) *GLUT10* mRNA expression in eWAT was analyzed by RT-PCR from *Glut10^{G128E}* mice and WT mice on a CD at 3 weeks of age (3W) or 20 weeks (20W) of age. n = 6 mice per group. (B-L) *Glut10^{G128E}* and WT

mice were analyzed at 3 weeks of age. (B-F) $Glut10^{G128E}$ mice have compromised eWAT development. (B) Tissue weight and (C) the triglyceride (TG) content of eWAT. n = 6 mice per group. (D) Hematoxylin and eosin (H&E) and Pref-1 immunohistochemistry staining of eWAT sections. (E) The cross-sectional area of adipose cells in eWAT is presented as the $log_2$ ratio of the area of cells in a particular size range in $Glut10^{G128E}$ mice relative to that of the same size range in WT mice. (F) The average adipocyte size in eWAT. E and F, n = 6 mice per group; more than 1000 adipocytes were analyzed in each mouse. (G -J) $Glut10^{G128E}$ mice have altered adipokine profiles. (G) The adipokine levels in serum of $Glut10^{G128E}$ and WT mice were determined by a mouse adipokine protein array. Serum samples from 8 mice per group were pooled. The arrowheads indicate signals with observable changes. (H) Quantification of relative levels of adipokines with observable changes. The average expression from two independent experiments and duplicate data in each array were analyzed separately. (I) Adipokine levels in serum were analyzed by ELISA. n = 4 mice per group. (J) mRNA expression of adipokines in eWAT was analyzed by RT-PCR. n = 6 mice per group. (K-L) $Glut10^{G128E}$ mice have reduced adipogenesis in eWAT. (K) The mRNA expression of adipogenesis-related genes in eWAT was determined by RT-PCR. n = 6 mice per group. (L) Protein expression levels were analyzed by western blotting. Protein samples from 6 mice per group were pooled. Data are shown as the mean ± SEM. $^*P < 0.05$, $^{**}P < 0.01$, $^{***}P < 0.001$.

WAT development, adipokines expression and energy metabolism (S7 Fig). We also noticed an increase in non-adipocyte cell population in $Glut10^{G128E}$ eWAT (Fig 3D). Furthermore, GLUT10 expression is higher in cells from the stromal vascular fraction of eWAT from both humans and mice [30], and its expression is readily detected in adipocyte precursor cells, including mouse embryonic fibroblasts (MEFs) and 3T3-L1 cells; of note, GLUT10 expression was not detected in immune cells (S8 Fig). We speculated that adipogenesis in $Glut10^{G128E}$ eWAT might be somehow impaired, so we examined adipogenesis in eWAT of WT and $Glut10^{G128E}$ mice at 3 weeks. Remarkably, immunohistochemical staining of preadipocyte factor 1 (Pref-1), a preadipocyte marker, was markedly increased in $Glut10^{G128E}$ eWAT (Fig 3D). Consistent with the increase in Pref-1 staining in $Glut10^{G128E}$ eWAT, both mRNA expression and protein levels of Pref-1were increased as well (Fig 3K and 3L); meanwhile, the expression levels of adipocyte-specific fatty acid binding protein 4 (FABP4) and key adipogenic transcription factors, including peroxisome proliferator-activated receptor gamma (PPARγ), PPARγ coactivator 1-alpha (PGC-1α) and CCAAT/enhancer-binding protein (C/EBPα), were all reduced in $Glut10^{G128E}$ eWAT (Fig 3K and 3L). These results suggest that adipogenic differentiation is impaired in $Glut10^{G128E}$ eWAT at an early age.

## GLUT10 deficiency cell-autonomously impairs adipogenesis and alters adipokine expression

To directly examine whether GLUT10 regulates adipogenesis, we used two *in vitro* adipogenic systems, MEFs and 3T3-L1 cells. MEFs were isolated from $Glut10^{G128E}$ and WT mice, while 3T3-L1 cells with stable knockdown of $Glut10$ expression ($shGlut10$) or stable expression of $shLuc$ (as a control) were used to test whether GLUT10 deficiency produces similar effects to the variant. As expected, adipogenic differentiation was much less successful in both $Glut10^{G128E}$ MEFs and $shGlut10$ 3T3-L1 cells, as evidenced by lower levels of lipid accumulation and triglyceride content (Fig 4A–4F). Consistent with its apparent role in adipogenesis, GLUT10 expression was high in preadipocytes and reduced after induction of adipogenic differentiation in 3T3L1 cells (Fig 4G). The expression levels of key adipogenic transcription factors (C/EBPα and PPARγ), adipocyte marker (FABP4), and a major adipokine (adiponectin), were all significantly reduced in $shGlut10$ 3T3-L1 cells during adipocyte differentiation (Fig 4H–4K). In contrast, the expression of inflammatory adipokines (IL-6 and TNF-α), was induced in $shGlut10$ 3T3-L1 cells (Fig 4L and 4M). These *in vitro* results agree well with our *in vivo* data (Figs 2 and 3), showing that GLUT10 regulates adipogenic differentiation and expression of adipokines.

## GLUT10 regulates intracellular ascorbic acid levels in preadipocytes and promotes adipogenesis

Next, we examined if the reduced adipogenic differentiation in GLUT10-deficient preadipocytes is due to low intracellular ascorbic acid levels. Ascorbic acid could be transported into

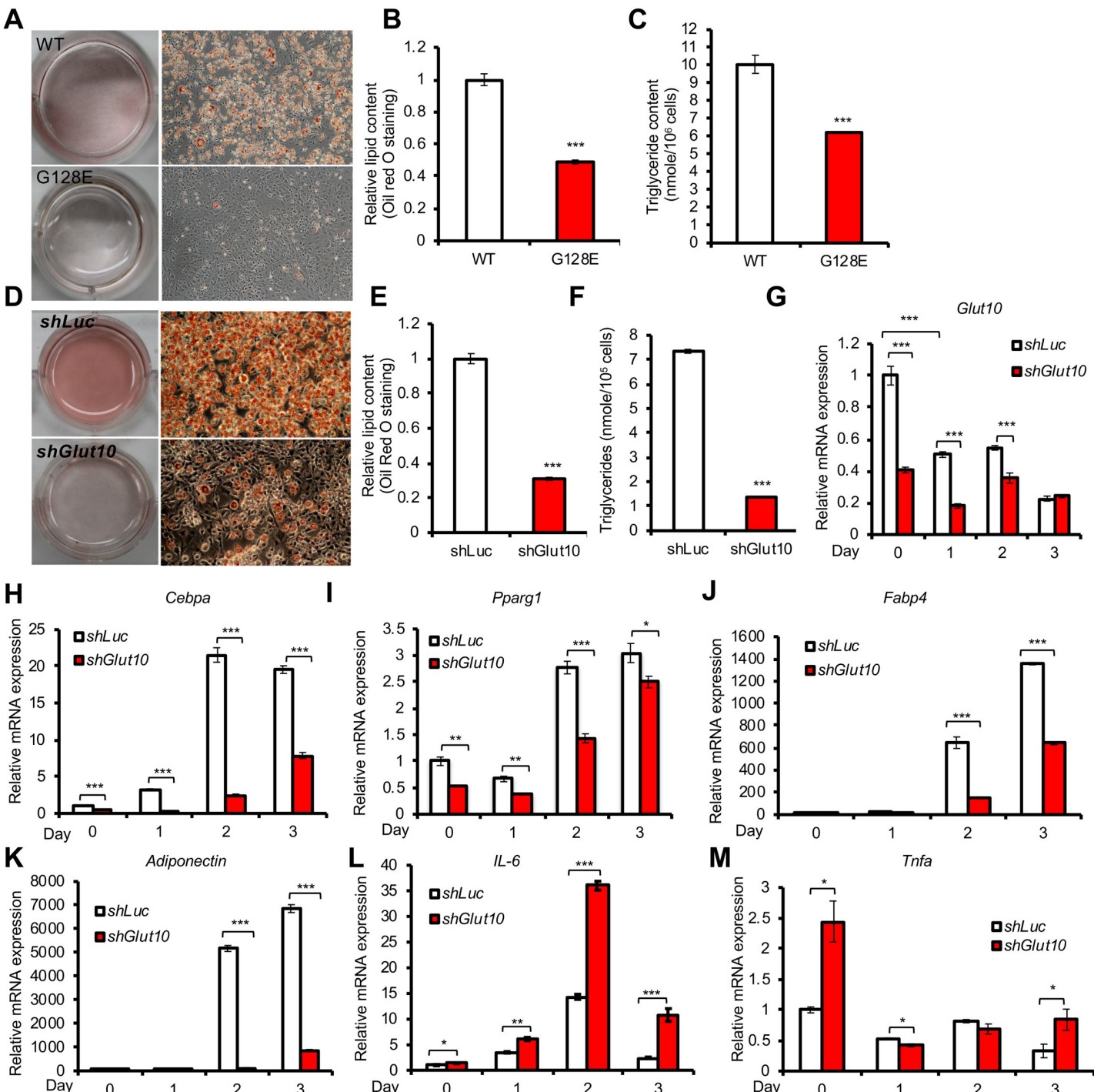

**Fig 4. GLUT10 deficiency cell-autonomously impairs adipogenesis and alters adipokine expression.** (A-C) MEFs were isolated from $Glut10^{G128E}$ and WT mice and induced adipogenic differentiation for 7 days. (A) Representative images show oil red O staining. (B) Quantification of oil red O staining and (C) intracellular triglyceride content (TG) are shown. (D-F) $Glut10$-knockdown ($shGlut10$) and control ($shLuc$) 3T3-L1 cells were induced for adipogenic differentiation for 6 days. (D) Representative images show oil red O staining. (E) Quantification of oil red O staining and (F) intracellular triglyceride (TG) content respectively. (G-M) The relative mRNA expression levels of $Glut10$ and indicated genes were determined by RT-PCR after induction of adipogenic differentiation for the indicated number of days. n = 3 independent experiments per group. Data are shown as the mean ± SEM. $^*P < 0.05$, $^{**}P < 0.01$, $^{***}P < 0.001$.

cells and intracellular compartment in its reduced form or oxidized form by SVCTs or GLUTs respectively. GLUT10 was expressed at relatively higher levels in 3T3-L1 preadipocytes compared to other expressed transporters (S9A Fig). Knockdown of GLUT10 expression did not alter the expression of other transporters (S9B Fig). As expected, DHA uptake and intracellular ascorbic acid levels were reduced in *shGlut10* 3T3-L1 cells cultured in 75 μM DHA (Fig 5A). Cultured the cells in a physiological concentration of ascorbic acid (75 μM) where ascorbic acid could be oxidized to DHA, the intracellular and nuclear ascorbic acid levels were also reduced in *shGlut10* 3T3-L1 cells (Fig 5B). These results demonstrate that GLUT10 maintains intracellular ascorbic acid levels in 3T3-L1 preadipocytes under physiological conditions. Supplementation with 75 μM ascorbic acid improved adipogenic differentiation in 3T3-L1 cells (Fig 5C and 5D). Furthermore, ascorbic acid supplementation had more pronounced effects in *shGlut10* 3T3-L1 cells (2-fold increase in lipid content) than *shLuc* 3T3-L1 cells (1.3-fold increase in lipid content) (Fig 5C and 5D). The same results were also validated in *Glut10*$^{G128E}$ MEFs (S10 Fig). Consistently, ascorbic acid supplementation induced the expression of key adipogenic genes, *Cebpa* and *Pparg*, in *shGlut10* 3T3-L1 cells (Fig 5E and 5F). These results suggest that GLUT10 is required for maintaining intracellular ascorbic acid level, and ascorbic acid positively regulates expression of *Cebpa* and *Pparg* and adipogenesis.

## GLUT10 maintains 5-hydroxymethylcytosine (5hmC) levels and gene expression of *Cebpa* and *Pparg*

We further explored the mechanism by which GLUT10 regulates ascorbic acid-mediated expression of *Cebpa* and *Pparg* and adipogenesis. Recent studies have shown that ascorbic acid serves as an enzyme cofactor for ten-eleven translocation (TET) dioxygenases, which catalyze 5-methylcytosine (5mC) demethylation to 5hmC and regulate cellular reprogramming and differentiation [42–46]. We therefore examined if GLUT10 affects ascorbic acid-dependent DNA demethylation. The *shGlut10* 3T3-L1 cells had lower global 5hmC levels than *shLuc* 3T3-L1 cells, and ascorbic acid supplementation increased global 5hmC levels (Fig 6A). We then directly investigated the DNA demethylation status in gene regions that are known to control expression of the central adipogenesis-regulating genes, *Cebpa* and *Pparg* [47, 48]. Importantly, ascorbic acid supplementation increased 5hmC levels in *Cebpa* and *Pparg* in 3T3-L1 cells (Fig 6B and 6C). These increased 5hmC levels in *Cebpa* and *Pparg* were positively correlated with the gene expression levels (Fig 5E and 5F), demonstrating that decreased intracellular ascorbic acid levels in *shGlut10* 3T3-L1 cells reduces DNA demethylation in *Cebpa* and *Pparg* and affects genes expression.

We further inhibited the ascorbic acid/iron/α-KG-dependent TET reaction to determine whether TET activity is necessary for proper adipogenesis. Cells were treated with the iron chelator desferrioxamine (DFO), the α-KG analog dimethyloxalylglycine (DMOG), or the chemical inhibitor 3-nitropropinic acid (NPA) to prevent succinate dehydrogenase A (SDHA) conversion of succinate to fumarate to inhibit TET enzyme activity (Fig 6D). DFO and DMOG both dose-dependently inhibited adipogenesis of *shGlut10* 3T3L1 cell (S11A–S11F Fig). Furthermore, treatment of cells with low doses of DMOG, DFO or NPA, which did not affect adipogenesis in control cells, was sufficient to significantly reduce adipogenesis in *shGlut10* 3T3-L1 cells (Fig 6E and 6F; S11 Fig). These results reveal that *shGlut10* 3T3-L1 cells were highly sensitive to DFO, DMOG or NPA-mediated inhibition of adipogenesis. These findings suggest that positive regulation of ascorbic acid by GLUT10 promotes TET enzyme activity to facilitate adipogenesis.

Based on the combined evidence from our *in vitro* models, we conclude that GLUT10 modulation of intracellular and nuclear ascorbic acid levels affects TET-mediated DNA

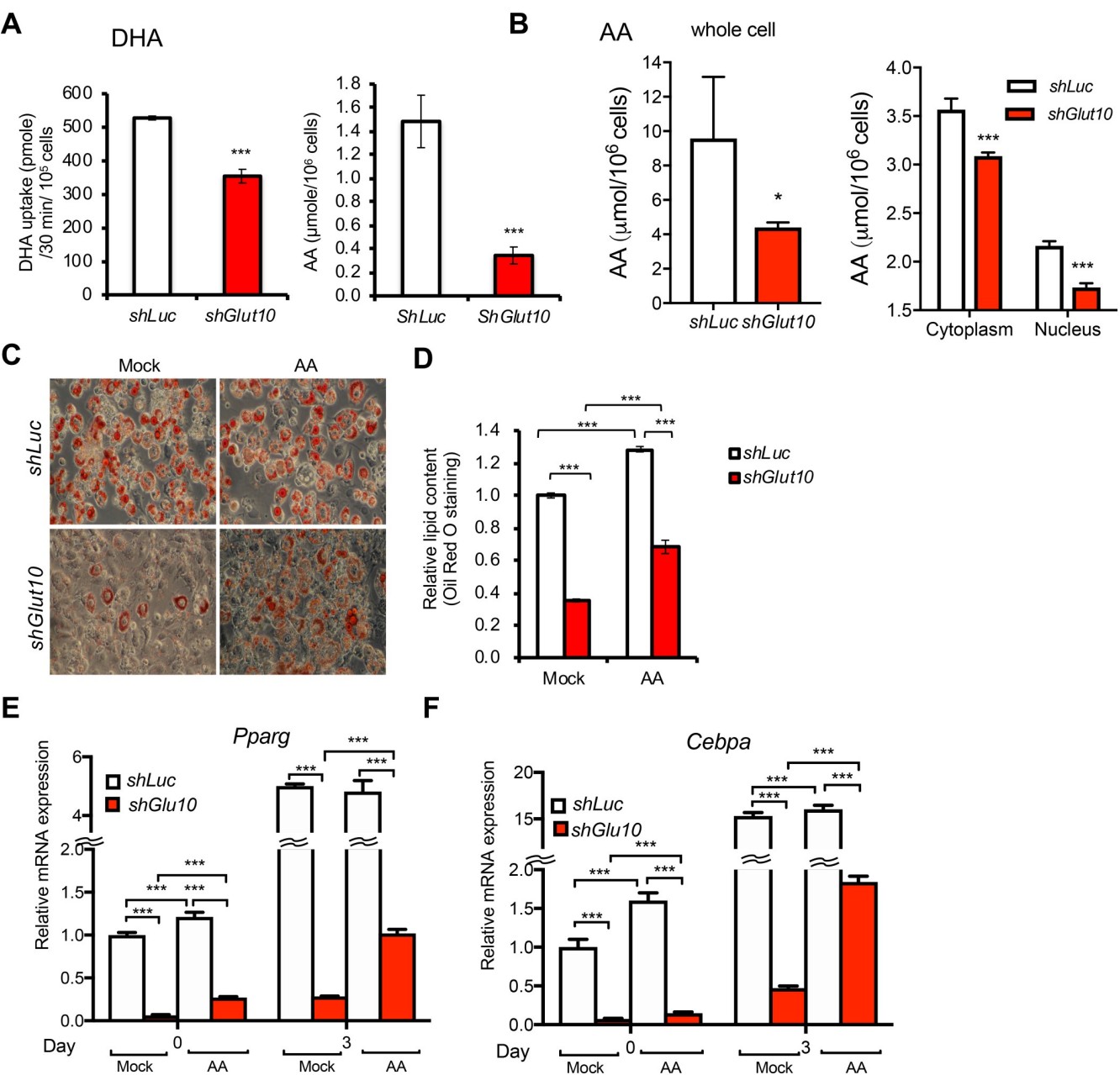

**Fig 5. GLUT10 modulates intracellular ascorbic acid levels and regulates adipogenesis.** (A) 3T3-L1 preadipocytes were incubated with 75 μM $^{14}$C-labeled DHA for 30 min. Intracellular $^{14}$C DHA levels (left panel) and intracellular ascorbic acid (AA) levels (right panel) were determined. (B) 3T3-L1 preadipocytes were cultured in medium containing 75 μM AA, and intracellular (left panel), cytoplasmic and nuclear AA levels (right panel) were determined. (C-F) *shGlut10* and *shLuc* 3T3-L1 cells were cultured in normal medium (4 μM AA from serum, mock) or supplemented with 75 μM AA for 2 days. Adipogenic differentiation was induced for 6 days in C and D. Adipogenic differentiation was induced for the indicated number of days in E and F. (C) Representative images of oil red O staining and (D) quantification of the oil red O staining. (E and F) Relative mRNA expression of *Cebpa* and *Pparg* was determined by RT-PCR. n = 3 independent experiments per group. Data are shown as the mean ± SEM. $^{*}P < 0.05$, $^{**}P < 0.01$, $^{***}P < 0.001$.

demethylation in *Cebpa* and *Pparg* genes, which regulates expression of the genes and adipogenesis (Fig 7). In *Glut10*$^{G128E}$ mice, the reduced adipogenesis contributes to reduced eWAT development, altered adipokine profile, and predisposal to HFD-induced inflammation and fibrosis of eWAT and severe metabolic consequences (Fig 7).

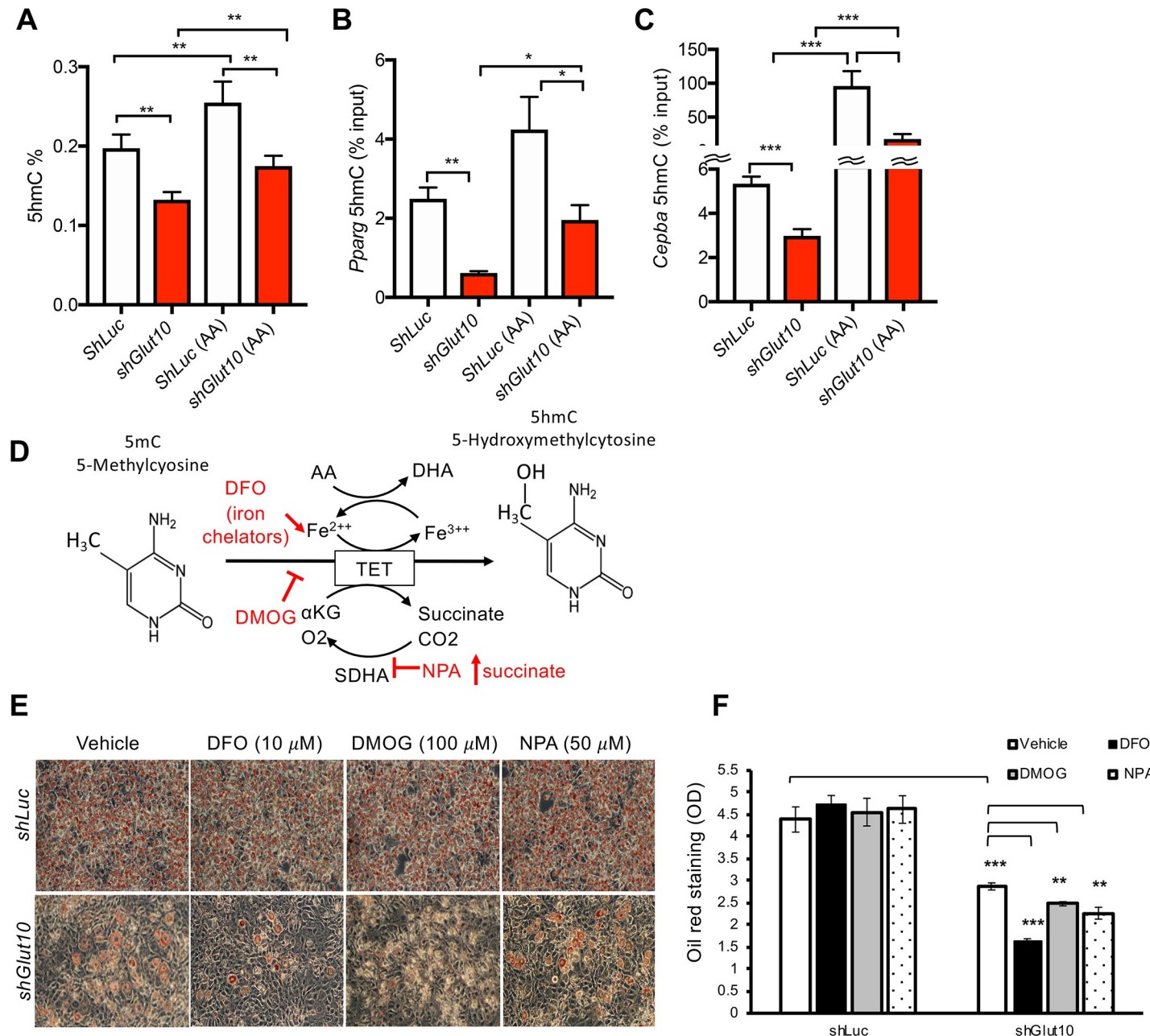

**Fig 6. GLUT10 modulates ascorbic acid-mediated DNA demethylation of *Cebpa* and *Pparg* and adipogenesis.** (A-C) *shGlut10* and *shLuc* 3T3-L1 cells were cultured in standard culture medium (4 μM AA from serum) or supplemented with 75 μM AA (AA). (A) Global 5hmC levels were quantified with an ELISA-based assay. (B and C) 5hmC levels in *Cebpa* and *Pparg* were quantified by 5hmC immunoprecipitation coupled with qPCR. (D) Schematic diagram depicting the inhibition of ascorbic acid (AA)-mediated TET DNA demethylation by DFO, DMOG or NPA. (E and F) *shGlut10* and *shLuc* 3T3-L1 cells were treated with either DFO (10 μM), DMOG (100 μM), NPA (50 μM), or a vehicle control along with 75 μM AA for 2 days and adipogenic differentiation was induced for 6 days. (E) Representative images of oil red O staining and (F) quantification of the oil red O staining. Data are shown as the mean ± SEM. n = 3 independent experiments per group. $^*P < 0.05$, $^{**}P < 0.01$, $^{***}P < 0.001$.

## Discussion

In this study, we demonstrate that GLUT10 plays an important role in WAT development and its loss of function contributes to HFD-induced metabolic dysregulation, providing new insights into the long-standing question of whether *SLC2A10* gene is associated with T2DM.

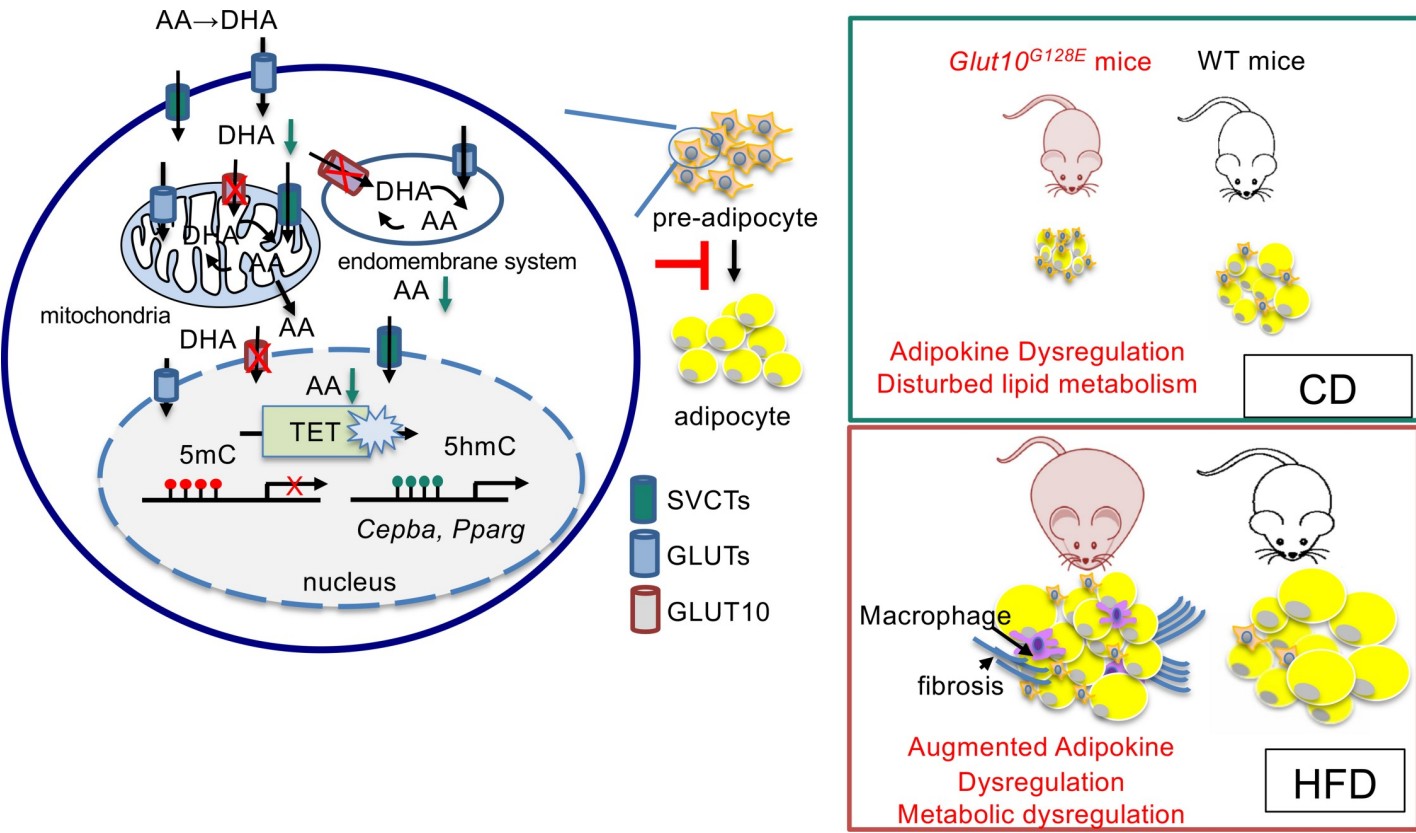

**Fig 7. Model depicts how GLUT10 modulates ascorbic acid levels to protect mice against HFD-induced metabolic dysregulation.** GLUT10 modulates ascorbic acid-mediated DNA demethylation and gene expression of *Pparg* and *Cebpa* to regulate adipogenesis, thereby affecting WAT development and metabolism.

Furthermore, our study provides an example of a genetic factor that affects WAT development to increase susceptibility toward diet-triggered T2DM in mice. We expect these findings will stimulate sophisticated population studies to probe gene-diet interactions in the development of T2DM, while also contributing key knowledge about disease prediction and prevention.

Large-scale GWAS have successfully uncovered more than 100 common variants associated with T2DM. However, these common variants explain only a small proportion of T2DM disease susceptibility and heritability, suggesting that rare variants and environmental interactions also contribute to T2DM risk [49]. Recent candidate gene approaches combined with functional studies have successfully identified rare variants with strong effects on T2DM risk [3–5]. Here, we identify a suggestive association between the *SLC2A10* locus and HbA1c levels in a non-diabetic Han Taiwanese population. While more than 700 rare variants (allele frequency < 1%) in *SLC2A10* gene are predicted to affect GLUT10 function (S2 Table and S3 Table), only a few have been demonstrated to cause ATS (S2 Table, labeled in red), and most variants are predicted to be benign or of uncertain significance. A large-scale study will therefore be needed to test the association of these rare variants with disease. Nevertheless, it is useful to perform experimental characterization of the functional effects of the potential variants. Here, we demonstrated that mice carrying the G128E variant have increased HbA1c levels, and they are predisposed to develop HFD-induced T2DM. We further demonstrated that adipogenesis is not only affected in *Glut10^{G128E}* eWAT and *Glut10^{G128E}* MEFs, but it is also affected in GLUT10 knockdown 3T3-L1 cells, suggesting other variants with compromised GLUT10 expression or function might have similar effects. Finally, although the contribution

of rare variants to overall disease burden in a population may be modest, the relatively large individual effect sizes may be useful for clinical risk prediction and disease prevention for certain individuals.

After *SLC2A10* was identified as the gene that causes ATS, patients with mild phenotypes were verified by sequencing the gene [50–52]. The life expectancy of ATS patients with mild phenotypes is longer than it is for those with severe phenotypes, and the connective tissue manifestations become more prominent with age [53]. Although no association with overt diabetes was observed in ATS patients [50], obesity was reported in some older ATS patients [54, 55], and one report mentioned that a carrier father developed hyperinsulinemia after rapid weight gain (10 kg in 2 years) [50]. Our findings suggest that follow-up studies monitoring blood glucose levels and diet control might be of benefit for ATS patients and carriers.

*Glut10*$^{G128E}$ mice have reduced adipogenesis, reduced WAT development, decreased adiponectin and increased IL-6 in eWAT at an early age. Reduced adipose tissue development (lipodystrophy) can lead to almost all the features of metabolic syndrome [56], and a subset of lipodystrophy patients exhibit reduced plasma adiponectin and numerous metabolic complications [57]. Adiponectin is predominantly secreted by adipocytes and plays various protective roles in the body, including anti-inflammatory, antidiabetic, antiatherogenic, and cardioprotective effects [58]. The *shGlut10* 3T3-L1 cells have reduced adipogenesis and reduced adiponectin levels. Previous reportes showed that serum levels of adiponectin are negatively correlated with visceral fat [59], and several lines of evidence showed that pro-inflammatory cytokines (i.e., TNF-α and IL-6) reduce adiponectin expression [60]. This reduced adiponectin level was observed prior to the onset of obesity and insulin resistance [60]. Thus, the reduced adiponectin in *Glut10*$^{G128E}$ mice might be due to reduced WAT and increased pro-inflammatory cytokines in *Glut10*$^{G128E}$ eWAT. Moreover, the dysregulation of adipokines in *Glut10*$^{G128E}$ mice might contribute to HFD-induced metabolic dysregulation. Notably, low-grade inflammation, especially increased IL-6, has been suggested to be an important factor to enhance dysregulation of fatty acid metabolism, adipocyte hypertrophy, immune cell infiltration, and the overproduction of extracellular matrix; by these actions, caloric excess promotes inflammation and fibrosis in eWAT [61–63]. The expression of lipogenesis and lipolysis genes was disturbed in eWAT of *Glut10*$^{G128E}$ mice on either a CD or HFD (S12A Fig), indicating a disturbance in lipid metabolism in *Glut10*$^{G128E}$ eWAT. This dysregulation of fatty acid metabolism may explain why the adipocyte size in *Glut10*$^{G128E}$ eWAT was reduced on CD and how it was increased to a level comparable to WT on HFD. Although sWAT was increased in *Glut10*$^{G128E}$ mice on HFD, no histological differences with WT were observed, suggesting the metabolic dysregulation in HFD-fed *Glut10*$^{G128E}$ mice might be mainly attributed to eWAT. To this point, significant differences in the functional effects of subcutaneous (sWAT) and visceral fat (eWAT) on metabolic dysfunction have been widely observed [64]. Together, these results and previous findings further support the idea that reduced adipogenesis, and adipokine dysregulation in eWAT of *Glut10*$^{G128E}$ mice predispose the mice to HFD-induced metabolic dysregulation.

In this work, we demonstrated GLUT10 modulates intracellular/nuclear ascorbic acid levels in 3T3L1 cells. Notably, decreased intracellular ascorbic acid levels was also osberved in MEFs isolated from *Glut10*$^{G128E}$ compared to WT controls (S10A Fig). Additionally, intracellular ascorbic acid levels were lower in freshly isolated SVF cells from *Glut10*$^{G128E}$ mice compared with SVF cells from WT mice (S13A Fig). Mice can endogenously synthesize ascorbic acid in the liver, and serum ascorbic acid levels were significantly increased in *Glut10*$^{G128E}$ mice on HFD (S13B Fig), suggesting an attempt at compensation for disrupted ascorbic acid homeostasis in *Glut10*$^{G128E}$ mice. The mechanisms contributing to this increased serum ascorbic acid levels in *Glut10*$^{G128E}$ mice are as yet unknown, however the observation supports the idea that GLUT10 deficiency disturbs ascorbic acid homeostasis in vivo.

We showed that GLUT10 modulates ascorbic acid-dependent DNA demethylation to regulate the expression of C/EBPα and PPARγ genes for adipogenic differentiation in 3T3L1 cells. Emerging evidence shows that ascorbic acid induces TET-dependent DNA demethylation to fine-tune gene expression and regulate cell reprograming [42, 43, 65]. Interestingly, ascorbic acid has been shown to enhance adipogenic differentiation by stimulating expression of C/EBPα and PPARγ at the initiation of adipogenesis though uncharacterized mechanisms [66]. However, the mechanisms governing intracellular/nuclear ascorbic acid levels and how might ascorbic acid might stimulate expression of C/EBPα and PPARγ are incompletely described. Our study provides evidence to support these previous findings and bridge some gaps in knowledge. Furthermore, C/EBPα and PPARγ are key transcription factors that orchestrate adipogenic and mitochondrial function [67]. Mitochondrial dysfunction decreases adipogenesis and increases expression of proinflammatory adipokines, such as IL-6, reducing the metabolic flexibility of WAT [68–70]. Indeed, the oxygen consumption rate (OCR), which reflects mitochondrial respiration, was significantly reduced in *shGlut10* 3T3L1 cells (S14A Fig). Ascorbic acid supplementation significantly improved OCR, while suppressing the increased expression of IL-6 and TNF-α in *shGlut10* 3T3L1 cells (S14B Fig). These results agree with previous studies and suggest that decreased C/EBPα and PPARγ expression and decreased mitochondrial function might increase the expression of IL-6 and TNF-α in *shGlut10* 3T3L1 cells. In addition, ascorbic acid is involved in diverse iron/α-KG dioxygenase reactions [13], which might also contribute to adipogenesis. Finally, in addition to aortic smooth muscle cells and WAT, GLUT10 expression can be detected in stomach and intestine in mice (S8 Fig). At present, we cannot exclude the possibility that GLUT10 deficiency in other tissues might affect adipogenesis through alternative mechanisms.

It is well established that ascorbic acid is an antioxidant/free radical scavenger. Hence, we explored the possibility that the antioxidant function of ascorbic acid may be important in adipogenesis. Consistent with the decreased intracellular ascorbic acid level, intracellular and mitochondrial reactive oxygen species (ROS) levels were increased in *shGlut10* 3T3-L1 cells (S15A and S15B Fig). Pre-incubating *shGlut10* 3T3-L1 cells with ascorbic acid or other antioxidants, NAC (*N*-acetyl-cysteine) and MitoQ (mitochondria-targeted coenzyme Q10), reduced intracellular and mitochondrial ROS levels (S15A and S15B Fig). Importantly, however, only ascorbic acid could enhance adipogenesis in *shGlut10* 3T3-L1 cells (S15C and S15D Fig). Furthermore, the intracellular and mitochondrial ROS levels were relatively low in *shLuc* 3T3-L1 cells, and while ascorbic acid supplementation did not further reduce the intracellular or mitochondrial ROS levels in these cells, it did promote adipogenesis (S15C and S15D Fig). These results suggest that ascorbic acid promotion of adipogenesis in 3T3L1 cells might not require its function as a free radical scavenger function. However, whether the disturbance of ascorbic acid homeostasis in *Glut10*^G128E mice might affect oxidative stress and inflammation to contribute to the eWAT phenotypes and metabolic consequences should be further explored.

In summary, we report previously uncharacterized roles of GLUT10 in adipogenesis and WAT development, which affect the sensitivity to HFD-induced metabolic dysregulation in mice. We also demonstrated that the *SLC2A10* gene is associated with intermediate traits of T2DM in a non-diabetic population. Thus, our study provides an example where a gene involved in WAT development affects sensitivity to HFD induced metabolic dysregulation.

## Materials and methods

### Ethics statement

The study was approved by the institutional review board and the ethics committee of Academia Sinica (AS-IRB01-101120), and informed consent was obtained from all participants in accordance with institutional requirements and the Declaration of Helsinki Principles.

## Association of *SLC2A10* polymorphisms with HbA1c level

We tested the association of the *SLC2A10* polymorphism with HbA1c in the Taiwan Super Control Study (TWSC) [35]. Subjects with diabetes were excluded, as defined by physician diagnosis, medication use, or HbA1c $\geq$ 6.5%. A total of 920 participants were included. The demographics of the participants were: age, 50 ± 17.8 (mean ± SD); 50% male; BMI, 23.6 ± 3.5 (mean ± SD); HbA11c, 5.2 ± 0.4 (mean ± SD). For HbA1c measurement, whole blood was collected from the participants, and HbA1c was analyzed by Cation Exchange HPLC TOSOH HIC-723 G7 (TOSH CORPRATION, Tokyo, Japan) and presented as the National Glycohemoglobin Standardization Program (NGSP) percent. SNPs were genotyped using Illumina Hap550duo chip with call rate $\geq$ 95%. The variants were imputed using post-QC genotypes and HapMap CHB+JPT (release no.22) as a reference panel. SNPs were analyzed for HbA1c association fitted to a linear regression model adjusted for sex and BMI.

## Mice

*Glut10*$^{G128E}$ mice on a C3HeB/FeJ background [36] were backcrossed to C57BL/6J for more than 10 generations, and animals on C57BL/6J background were used in this study. Mice were housed in a controlled environment with a 14-h light/10-h dark cycle at 21–23˚C under specific pathogen-free conditions. Standard rodent diet (CD) contained 13% energy from fat (LabDiet 5010 rodent Diet, PMI Nutrition International Inc., Brentwood, MO, USA), and the high-fat diet (HFD) contained 60% energy from fat (58Y1, Young Li Trading Co., New Taipei City, Taiwan). All animal protocols were approved by the Institutional Animal Care and Utilization Committee, Academia Sinica (Protocol #14-12-795). Male mice were used in this study. No data were excluded in the analyses.

## Glucose and insulin tolerance test (GTT and ITT)

For fasting blood glucose measurement, blood samples were collected from mice after overnight fasting. For the GTT, mice at 16 weeks of age were fasted for 18 h and then given an intraperitoneal injection of glucose (2 g/kg). For the ITT, mice at 18 weeks of age were fasted for 8 h and administered an intraperitoneal injection of insulin (0.75 U/kg, Humulin R U100, Lilly, Eli and Company, Indianapolis, IN, USA). Blood samples were collected 15, 30, 45, 60, 75, and 90 min post-injection. Blood glucose levels were assessed using a glucometer (Accu-Chek Performa, Roche Medical Diagnostic Equipment Co., Taiwan).

## Body composition

Mouse body composition was analyzed by Bruker's minispec LF50 Body Composition Analyzers in the Taiwan Mouse Clinic at Academia Sinica.

## Histological analysis and immunohistochemistry

Tissue sections were stained with hematoxylin and eosin (H&E) or Masson's trichrome stain. For immunohistochemistry, deparaffinized tissue sections were incubated with primary and secondary antibodies, as indicated in Supplementary Data.

## Triglyceride content

Adipocytes or minced tissues were lysed in lysis buffer (5% NP-40 in H$_2$O). The lysis solutions were slowly heated to 80–100˚C for 2–5 min to ensure triglycerides were dissolved. Total triglyceride content was determined with a Triglyceride Quantification Colorimetric/Fluorometric kit (BioVision Inc, Milpitas, CA, USA).

## Blood chemistry and adipokine assays

Blood was collected from cardiac puncture for blood chemistry and adipokine assays.

The enzyme activities of GOT and GPT, in addition to TCHO and TG levels were analyzed from serum samples using Fuji biochemical slides and a Fuji Dri-Chem 4000i analyzer (Fuji-film Cooperation, Taipei, Taiwan) in the Taiwan Mouse Clinic at Academia Sinica. Plasma levels of adiponectin, leptin, IL-6, and insulin were measured using mouse ELISA kits (Merck Millipore, Taipei, Taiwan). The plasma free fatty acid was measured using an ELISA kit (ab65341, Abcam, Cambridge, MA, USA), and blood HbA1c was measured using the mouse Hemoglobin A1c (HbA1c) Assay kit (Crystal Chem, Elk Grove Village, IL, USA).

## Quantitative reverse transcriptase polymerase chain reaction (qRT-PCR)

Total RNA from cells or tissues was isolated, converted to cDNA, and then subjected to qRT-PCR as described previously [20]. Expression data were normalized to 36B4 mRNA in adipose tissues or β-actin mRNA levels in liver tissues. Gene-specific primer sequences are provided in S1 File.

## Western blot

Total protein lysates from tissues were used for analysis. After Western blotting, protein levels were detected using enhanced chemiluminescence (Millipore Merck, Taipei, Taiwan). The primary and secondary antibodies are listed in S1 File.

## Adipokine array

Mice were fasted for 18 h before their blood was collected. Serum was isolated, and 100 μL was used in a commercial adipokine array (ARY-013, R&D systems, Minneapolis, MN, USA) according to the manufacturer's protocol. The proteome profiler adipokine array detects 38 adipokines in duplicate on nitrocellulose membranes as described in the Supplementary Data.

## Cell culture, adipocyte differentiation, transfection, and stably transfected cell lines

3T3-L1 preadipocytes were obtained from the Taiwan Bioresource Collection and Research Center (Hsinchu, Taiwan). Generation of the Glut10-knockdown and control stable cell lines was accomplished using *pLKO* shRNA vectors encoding shRNAs targeting *Glut10* or luciferase sequences as previously described [20]. MEFs were isolated from *post coitus* day 13.5 *Glut10*$^{G128E}$ and WT embryos as previously described [71]. All MEFs in this study were used within two passages. Cells were cultured and differentiation was induced for the indicated number of days as described previously [16].

## Oil Red O staining and quantification

Cells were fixed with 10% formaldehyde at the indicated times after induction of differentiation and were stained with 0.3% Oil Red O (Sigma-Aldrich Inc., St. Louis, MO, USA) in 60% isopropanol. The Oil Red O was dissolved in isopropanol for quantification by absorbance spectrophotometry at 510 nm.

## DHA uptake and intracellular AA measurements

DHA uptake was determined using $^{14}$C-labeled DHA, and intracellular AA levels were determined using an Ascorbic Acid Assay Kit (Abcam, Cambridge, England, UK).

## Global 5hmC quantitation

Genomic DNA was purified and sonicated. Global DNA hydroxymethylation (5hmC) was measured using commercial kits (ab117131, Abcam). The percentage of 5hmC in each sample was calculated. The calibration curve and negative controls were measured in duplicate, while the samples were measured in triplicate.

## Gene-specific 5hmC quantitation

Genomic DNA was purified and sonicated. The DNA fragments containing 5hmC were enriched and isolated using commercial kits (ab117134, Abcam), and qPCR was used to quantify the enrichment of 5hmC DNA at specific gene loci. The sequences of primer pairs are listed in the Supplementary Data.

## Statistical analysis

A two-tailed Student's $t$-test was used to test for differences between groups. $P$-values less than 0.05 were considered statistically significant.

## Supporting information

**S1 File. Supplementary materials and methods.**
(PDF)

**S1 Table. SNPs in *SLC2A10* region associated with BMI, glucose tolerance, insulin resistance or T2D-related phenotypes in independent human populations and studies.**
(PDF)

**S2 Table. The potential variants in SLC2A10 region that might affect the GLUT10 expression or function.**
(XLSX)

**S3 Table. Variants in the *SLC2A10* region**
(PDF)

**S1 Fig. GLUT10<sup>G128</sup> is conserved during evolution, and the GLUT10<sup>G128E</sup> variant is predicted to be benign.** (A) Schematic model of the structure of GLUT10. (B) Amino acid sequence alignment of GLUT10 in *Homo sapiens*, *Pan troglodytes*, *Macaca mulatta*, *Canis lupus familiaris*, *Bos taurus*, *Mus musculus* and *Rattus norvegicus*. The yellow highlight indicates the G128 residue is highly conserved in mammals. The conserved domain is indicated in red. The transmembrane domains (TM) 3, 4 and 5 are indicated. (C) The variant effect predicted by Ensembl Variant Effect Predicator. The variant is predicted to be benign.
(PDF)

**S2 Fig. *Glut10<sup>G128E</sup>* mice exhibit no significant difference in food intake, physical activity, or energy expenditure.** WT and *Glut10<sup>G128E</sup>* mice were fed a CD or HFD from 5 to 20 weeks of age. Data were collected from mice at the conclusion of feeding. (A) Average food intake over a 24-h period. (B) Physical activity over a 24-h period; the bar below the graph indicates the light and dark portion of the day. Physical activity was measured in WT and *Glut10<sup>G128E</sup>* mice under a HFD using Clever Sys HomeCageScan TM3.0. (C–F) The metabolic indicators at different temperatures over a 79-h period. (C) Consumption of O2, (D) CO2 production, (E) Respiratory Exchange Ratio (RER- an assessment of the metabolic exchange of oxygen for carbon dioxide) = $VCO_2/VO_2$, and (F) heat generation were measured by a comprehensive laboratory animal monitoring system (CLAMS). Bar graphs on the left indicate the average

values during the light and dark cycle at different temperatures. Error bars, SEM. n = 8 mice in each group.
(PDF)

**S3 Fig. HFD-fed *Glut10*^G128E^ mice have impaired insulin-induced AKT phosphorylation in insulin-responsive tissues.** *Glut10* and WT mice fed a HFD from 5 to 20 weeks of age were fasted overnight and injected with saline (Mock) or insulin (0.5 U/kg). Mice were killed 30 min after injection. Insulin-induced AKT phosphorylation in (A) eWAT, (B) liver, and (C) skeletal muscle was analyzed by western blotting; relative intensity was quantified (right panels). Error bars, SEM. n = 3 mice in each group.
(PDF)

**S4 Fig. The appearance and histology of sWAT, iBAT, and pancreatic tissue of WT and *Glut10*^G128E^ mice after CD or HFD feeding.** *Glut10*^G128E^ mice and WT mice were fed a CD or HFD from 5 to 20 weeks of age, after which the mice were killed and tissues collected for analysis. (A) Representative photographs of mice and their tissues. Each square on the green mat is 1 cm × 1cm. (B-D) H&E staining of iBAT, sWAT, and pancreas sections.
(PDF)

**S5 Fig. *Glut10*^G128E^ mice on a CD or HFD exhibit increased inflammation and adipokine dysregulation in eWAT.** *Glut10*^G128E^ mice and WT mice were fed with a normal diet (CD) or HFD from 5 to 20 weeks of age. Data were analyzed from the mice at the conclusion of feeding. mRNA expression levels were analyzed in eWAT by RT-PCR. n = 6 mice per group. (A) Genes involved in inflammation and fibrosis. (B) *Adiponectin* and *leptin* expression. (C) *Glut10* expression. Data are shown as the mean ± SEM. $^*P < 0.05$, $^{**}P < 0.01$, $^{***}P < 0.001$.
(PDF)

**S6 Fig. *Glut10*^G128E^ mice have no obvious abnormalities in vasculature on a CD or HFD.** *Glut10*^G128E^ mice and WT mice were fed a CD or HFD from 5 to 20 weeks of age. Data were collected from the mice at the conclusion of feeding. (A) H&E staining of aorta sections. (B) Fluorescence staining of vascular tissues in eWAT sections. Fluorescence microscopy of eWAT after perfusion with fluorescent microbeads revealed similar vascularization in WT and *Glut10*^G128E^ mice under both CD and HFD conditions.
(PDF)

**S7 Fig. Analysis of dysregulated genes in eWAT of GLUT10 variant mice.** The gene expression profiles in eWATs from WT and *Glut10*^G128E^ mice at 3 weeks were analyzed by RNA-seq with cutoffs of fold change (FC) ≥ 1.5 and ≤ 0.5. Samples from 8 mice per group were pooled. (A) Pie chart shows 395 genes were upregulated and 229 genes were downregulated in eWAT of *Glut10*^G128E^ mice. (B and C) The dysregulated genes were analyzed by ingenuity pathway analysis (IPA) for annotations of disease and biological function and upstream regulators. The categories of diseases and bio function showed many dysregulated genes are involved in adipocyte differentiation, development of adipose tissue, lipid metabolism and glucose metabolism. The upstream regulator analysis identified FGF21 (fibroblast growth factor 21, a metabolic pathway regulator), LEPR (leptin receptor, involved in the regulation of body weight), LEP (leptin, involved in the regulation of body weight), PPARGC1A (peroxisome proliferator- activated receptor gamma coactivator 1-alpha, PGC-1α, involved in adipogenesis and energy metabolism), and PPARγ (peroxisome proliferator-activated receptor gamma, a key adipogenic transcription factor) as the top 5 upstream regulators for the dataset. These analyses suggest that GLUT10 regulates adipogenesis, WAT development, adipokine expression and energy metabolism.
(PDF)

**S8 Fig. Relative GLUT10 expression in human and mouse tissues.** The GDS596 (human) and GDS592 (mouse) expression data sets, available on the NCBI GEO database from Su et al. (https://www.ncbi.nlm.nih.gov/geo/query/acc.cgi?acc=GSE1133), were reanalyzed by BioGPS (http://biogps.org). Relative signal intensity values are depicted in arbitrary units with ranges. The red arrows indicate tissues examined in the current study.
(PDF)

**S9 Fig. The expression of SVCTs and GLUTs in *shLuc* and *shGlut10* 3T3-L1 cells.** The mRNA expression levels in *shLuc* and *shGlut10* 3T3-L1 cells were analyzed by RT-PCR. (A) Only the expression of GLUT10, GLUT1, GLUT8 and SVCT 2 can be detected in 3T3-L1 cells. The relative expression of the expressed transporters was shown in dCT values normalized to β-actin expression. (B) The changes of gene expression in *shGlut10* 3T3-L1 cells were compared to *shLuc* 3T3-L1 cells. n = 3 independent experiments. Data are shown as the mean ± SEM. $^*P < 0.05$, $^{**}P < 0.01$, $^{***}P < 0.001$.
(PDF)

**S10 Fig. Ascorbic acid rescues the adipogenesis ability in *Glut10*$^{G128E}$ MEFs.** (A) The intracellular ascorbic acid (AA) levels. The MEFs were cultured in medium with 75 μM AA supplemented. (B and C) The MEFs were treated with 75 μM AA or vehicle control (Mock) for 2 days and induced for adipogenic differentiation for 7 days. (B) Representative images of oil red O staining are shown. (C) Quantification of the oil red staining. The data represent mean ± SEM, n = 3 independent experiments per group. $^*p < 0.05$, $^{**}p < 0.01$, $^{***}p < 0.001$.
(PDF)

**S11 Fig. GLUT10-deficient 3T3-L1 cells are more sensitive to DFO and DMOG inhibiting adipogenesis.** *shGlut10* and *shLuc* 3T3-L1 cells were pretreated with the indicated dose of drug along with vitamin C for 2 days, and adipogenic differentiation was induced for 6 days. (A-C) Cells were pretreated with DMOG or (D-F) DFO. (A, D) Representative images of oil red O staining are shown. (B, C, E, F) Quantification of oil red O staining is shown as the (B, E) OD value and (C, F) relative inhibition rate. Data are shown as the mean ± SEM. n = 3 independent experiments per group. $^*P < 0.05$, $^{**}P < 0.01$, $^{***}P < 0.001$.
(PDF)

**S12 Fig. *Glut10*$^{G128E}$ mice on a CD or HFD exhibit disrupted lipid metabolism and adipogenesis in eWAT.** *Glut10*$^{G128E}$ mice and WT mice were fed with a normal diet (CD) or HFD from 5 to 20 weeks of age. Data were analyzed from the mice at the conclusion of feeding. (A and B) mRNA expression levels were analyzed in eWAT by RT-PCR. n = 6 mice per group. (A) Genes involved in lipogenesis and lipolysis. Data are shown as the mean ± SEM. $^*P < 0.05$, $^{**}P < 0.01$, $^{***}P < 0.001$.
(PDF)

**S13 Fig. Ascorbic acid homeostasis is disturbed in *Glut10*$^{G128E}$ mice.** (A) Intracellular ascorbic acid (AA) levels were determined in SVF cells from eWATs of WT and *Glut10*$^{G128E}$ mice. The eWATs from 7 male *Glut10*$^{G128E}$ mice and 6 male WT mice (at age 6–8 weeks) were freshly isolated and pooled for AA measurement. (B) Serum ascorbic acid levels were measured from *Glut10*$^{G128E}$ mice and WT mice fed a CD or HFD from 5 to 20 weeks of age.
(PDF)

**S14 Fig. Ascorbic acid supplementation improves mitochondrial function and reduces inflammatory cytokine expression in *shGlut10*3T3-L1 preadipocytes.** The *shGlut10* and *shLuc* 3T3-L1 preadipocytes were pretreated with 75 μM vitamin C (AA) or vehicle control (Mock) for 2 days. (A) OCR was measured by a Seahorse Bioanalyzer. (B) The mRNA

expression of Il-6 and Tnf-α were determined by RT-PCR. Data are shown as the mean ± SEM. $^*P < 0.05$, $^{**}P < 0.01$, $^{***}P < 0.001$. n = 5 per group in A; n = 3 per group in B. $^*P < 0.05$. $^{**}P < 0.01$, $^{***}P < 0.001$.
(PDF)

**S15 Fig. Ascorbic acid enhances adipogenesis in *shGlut10* 3T3-L1 cells but other antioxidants do not.** 3T3-L1 cells were pretreated with 75 μM ascorbic acid (AA), 2 mM N-Acetyl-Cysteine (NAC), 50 nM mitochondria-targeted coenzyme Q10 (MitoQ) or vehicle control (Mock) for 2 days and induced for adipogenic differentiation. (A) Intracellular ROS levels and (B) mitochondrial ROS levels were determined in 3T3-L1 preadipocytes after 2 days of treatment. (C) The representative images show oil-red O staining and (D) Oil-red O staining was quantified in 3T3-L1 cells treated for 2 days and after induction of adipogenic differentiation for 6 days. n = 3 per group.
(PDF)

**S1 Data. Underlying numerical data corresponding to the main figures.** Spreadsheet containing all raw data used to generate graphs in the main figures.
(XLSX)

## Acknowledgments

We thank the Taiwan Mouse Clinic funded by the Ministry of Science and Technology (MOST) of Taiwan for their technical support in the determination of blood chemistry, home cage and body composition. We thank Marcus Calkins, ICOB, Academia Sinica, Taiwan for English editing.

## Author Contributions

**Conceptualization:** Yi-Ching Lee.

**Data curation:** Chung-Lin Jiang, Wei-Ping Jen, Chang-Yu Tsao, Li-Ching Chang, Chien-Hsiun Chen, Yi-Ching Lee.

**Formal analysis:** Li-Ching Chang, Chien-Hsiun Chen, Yi-Ching Lee.

**Funding acquisition:** Yi-Ching Lee.

**Investigation:** Chung-Lin Jiang, Yi-Ching Lee.

**Methodology:** Yi-Ching Lee.

**Project administration:** Yi-Ching Lee.

**Resources:** Yi-Ching Lee.

**Supervision:** Yi-Ching Lee.

**Validation:** Yi-Ching Lee.

**Visualization:** Yi-Ching Lee.

**Writing – original draft:** Yi-Ching Lee.

**Writing – review & editing:** Chung-Lin Jiang, Wei-Ping Jen, Li-Ching Chang, Chien-Hsiun Chen, Yi-Ching Lee.

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
