## [Decision Letter · Decision Letter 0]

22 Feb 2020

Dear Dr Lee,

Thank you very much for submitting your Research Article entitled 'Glucose Transporter 10 Modulates Adipogenesis via Vitamin C-Mediated Pathway Contributing to Diet-Induced Metabolic Dysregulation' to PLOS Genetics.

The manuscript was fully evaluated at the editorial level and by two independent peer reviewers. As you will see, both reviewers are generally positive, but express a series of concerns that would need to be addressed with additional experiments to move forward. Based on the reviews, we will not be able to accept this version of the manuscript, but we would be willing to review again a much-revised version. We cannot, of course, promise publication at that time.

If you decide to revise the manuscript for further consideration at PLOS Genetics, please aim to resubmit within the next 60 days, unless it will take extra time to address the concerns of the reviewers, in which case we would appreciate an expected resubmission date by email to plosgenetics@plos.org.

[LINK]

We are sorry that we cannot be more positive about your manuscript at this stage. Please do not hesitate to contact us if you have any concerns or questions.

Yours sincerely,

Gregory S. Barsh

Editor-in-Chief

PLOS Genetics

Gregory Copenhaver

Editor-in-Chief

PLOS Genetics

Reviewer's Responses to Questions

**Comments to the Authors:**

Reviewer #1: The authors claim that GLUT10 mediates adipogenesis by vitamin-C dependent demethylation of DNA. They employed a previously established Glut10^G128E mouse strain which carries a human rare variant with compromised GLUT10 function and revealed that the mice show HFD-induced metabolic disorders such as obesity and insulin resistance. Their epidydimal WAT showed reduced weight and increased inflammation and fibrosis, which may cause increased fat storage in the subcutaneous WAT and ectopic lipid accumulation in liver and BAT. Authors also showed that GLUT10 cell-autonomously mediates adipocyte differentiation by presenting that shGlut10 3T3-L1 cells showed reduced adipogenesis as well as reduced uptake of vitamin C. In addition, supplementation of vitamin C induced adipocyte differentiation by mediating mRNA expression which was associated with induced 5hmC levels in the genomic regions of Cebpa and Pparg. Together, the authors conclude that GLUT10 mediates vitamin-C mediated DNA demethylation and gene expression of Cebpa and Pparg to affect adipogenesis. Furthermore, the authors present a clinical link that the SLC2A10 gene which encodes GLUT10 protein is associated with type 2 diabetes mellitus-related phenotypes in non-diabetic Han Taiwanese. The authors’ claim is overall convincing, and their findings are very interesting. However, there are several concerns to be addressed as listed.

Points to be addressed

1. Table S5 is missing.

2. Several figure numbers in the manuscript are incorrectly indicated (i.e. Figure S3A in line 205, Figure S3B in lines 206-207, Figure S3A in line 212). All figure numbers should be precisely indicated.

3. Authors claim in lines 136-139 that “on a HFD, Glut10^G128E mice gained more weight than WT mice, despite both genotypes exhibiting comparable food intake, physical activity, energy expenditure, and respiratory exchange ratio (RER) on either a CD or HFD”. However, HFD-treated Glut10^G128E mice show a clear reduction trend of VO2, VCO2 and heat production compared to other groups in Figure S2. It is suggested to state more precisely that the reduced trend of energy expenditure in HFD-treated Glut10G128E mice would be a cause of their obesity.

4. While it is reported that the serum levels of adiponectin are inversely correlated with visceral fat amount, Glut10^G128E mouse shows reduced eWAT with low serum adiponectin level. Authors should discuss the possible mechanisms. For example, if it is considered that the fat accumulation in scWAT, BAT, and ectopic organs affect the phenotype. It is also suggested to discuss the mechanism that GLUT10 and ascorbic acid mainly target eWAT.

Reviewer #2: In this manuscript, the authors describe the effects of modulating GLUT10 expression or activity and the response in adipose tissue. These studies implicate some effects of GLUT10 in energy metabolism. However, some of the conclusions reached by the authors are reached through implied effects of ascorbic acid in this system, and are poorly supported by their direct line of questioning. It is recommended that additional data is needed to refer to this phenomenon as 'vitamin C-mediated' as mentioned in the title. A list of specific points that need to be addressed in this manuscript follows.

Major Issues:

1. GLUT10 is present on endomembranes, as the authors note on P13, rather than the plasma membrane. It is unclear how a disruption of this protein results in whole-cell changes in vitamin C transport systems. Cells can transport DHA though various members of GLUT family - many of which are present in 3T3 cells. Thus, the authors should survey all other GLUT members involved in DHA transport to determine if shGlut10 3T3-L1 cells show aberrations in those systems. In other words, is the disruption of GLUT10 shRNA specific?

2. Ascorbic acid is transported through SVCT2 in 3T3-L1 cells, yet no attempt was made to determine SVCT2 levels in these cells, especially in the shGlut10 vs. controls. These are critical experiments (along with those in point #1) to determine if this phenomenon is truly due to changes in GLUT10

3. All of the experiments concerning vitamin C are conducted in 3T3-L1 cells, and no evaluation of the effects of GLUT10 disruption on vitamin C levels, vitamin C transport, or glucose transport were made in the cells isolated from transgenic animals. These are also critical to understanding the effects of GLUT10 disruption, and making comparison from a cell culture model and in vivo disruption. Cells placed in culture are maintained without ascorbic acid and the effects of its addition are unpredictable - in vivo experiments are needed to define physiologically relevant changes.

4. In Fig S4E, serum ascorbate levels in G128E mice appear to be higher than than WT in both feeding conditions, but only significantly higher in the high-fat diet. This would imply that GLUT10 is involved in the biosynthesis of ascorbic acid in the rodent liver or the retention of ascorbic acid in the kidney - yet neither of these phenomenon are discussed in the paper.

Minor Issues:

1. Ascorbic acid is not a vitamin for rodents. Since no human cells were used in these experiments, it would be proper to use it chemical name.

2. In several parts of the paper, the authors suggest that vitamin C contributes to T2DM, without defining the exact relationship. In some places the wording implies that high vitamin C would promote diabetes, but it the opposite is more likely to be true.

3. The reason that vitamin C is inversely related to body fat composition is due to the increased amount of inflammation/oxidative stress that obesity brings. There is no credible source that suggests vitamin C can reverse obesity alone, but it may mitigate some of its effects.

4. The relevant text about GLUT10 being present in endomembranes should be part of the introduction, including text about the other transport mechanisms for vitamin C.

5. When the authors state that GLUT10 is associated with diabetes phenotypes, it is unclear if the authors are speaking of normal GLUT10 activity or altered/SNP activity or expression.

6. On Page 5, line 111-12, the authors imply that GLUT10 controls vitamin C levels in cells - but these results have only been shown in cell culture and show no data from animal or humans to back up this claim.

7. Ascorbic acid has effects on 3T3 differentiation that are independent of GLUT10 - it is unclear if the authors have properly attributed changes in gene expression (and 5hmC) with AA supplementation. These should be conducted on cells that have been fully differentiated.

8. The effects of AA in cell culture can span both pro-oxidant and anti-oxidant effects that are not always apparent in animal systems. These are typically thoughts of as cell culture artifacts due to the high oxygen and iron content in cell culture systems. Besides performing these experiments in isolated issues from animals, it is suggested that the authors attempt to determine if changes in vitamin C levels are reducing or increasing ROS burden. Also, the increased ROS production by GLUT10 changes (or other glucose regulation) could be responsible for these apparent effects on vitamin C, and should be explored further.

**Have all data underlying the figures and results presented in the manuscript been provided?**

Reviewer #1: No: Table S5 is missing.

Reviewer #2: Yes

PLOS authors have the option to publish the peer review history of their article (what does this mean?). If published, this will include your full peer review and any attached files.

Reviewer #1: No

Reviewer #2: Yes: Alexander J Michels

---

## [Decision Letter · Decision Letter 1]

27 Apr 2020

Dear Dr Lee,

Thank you very much for submitting your Research Article entitled 'Glucose transporter 10 modulates adipogenesis via an ascorbic acid-mediated pathway to protect mice against diet-induced metabolic dysregulation' to PLOS Genetics.

The manuscript was seen by the two original peer reviewers. As you will see, both reviewers are positive; however, there are some remaining minor concerns from reviewer #2 that we ask you address in a hopefully final revision that will not necessarily require additional external review. 

We therefore ask you to modify the manuscript according to the review recommendations before we can consider your manuscript for acceptance. Your revisions should address the specific points made by each reviewer.

[LINK]

Yours sincerely,

Gregory S. Barsh

Editor-in-Chief

PLOS Genetics

Gregory Copenhaver

Editor-in-Chief

PLOS Genetics

Reviewer's Responses to Questions

**Comments to the Authors:**

Reviewer #1: The authors have adequately addressed my previous comments. In my view, this manuscript is suitable for publication in PLoS Genetics.

Reviewer #2: Although this manuscript is improved, there is still a couple of remaining issues with vitamin C transport that need clarification.

1. In their response, the authors claim that other vitamin C transport proteins (GLUTs and SVCT2) are not affected by the knockdown of GLUT10. This data, however, is not provided. Given the wealth of other data that the authors provided, this omission is notable. Can this be included?

2. The data with AA uptake by the cells in Figure 5B appears to indicate that the GLUT10 knockdown cells are unable to transport AA at the same rate they did in the WT cells. However, given that this experiment was conducted over a time and cell culture condition where ascorbate could oxidize to DHA, this could be attributable to changes in DHA transport alone. Either the Figure and the corresponding text needs to be changed to altered to reflect this, or the experiment needs to be repeated under conditions where AA transport alone is measured.

**Have all data underlying the figures and results presented in the manuscript been provided?**

Reviewer #1: No: The RNA sequencing data should be deposited within a public data repository.

Reviewer #2: Yes

PLOS authors have the option to publish the peer review history of their article (what does this mean?). If published, this will include your full peer review and any attached files.

Reviewer #1: Yes: Takeshi Inagaki

Reviewer #2: Yes: Alexander J Michels

---

## [Editor Report · Decision Letter 2]

2 May 2020

Dear Dr Lee,

We are pleased to inform you that your manuscript entitled "Glucose transporter 10 modulates adipogenesis via an ascorbic acid-mediated pathway to protect mice against diet-induced metabolic dysregulation" has been editorially accepted for publication in PLOS Genetics. Congratulations!

Yours sincerely,

Gregory S. Barsh

Editor-in-Chief

PLOS Genetics

Gregory Copenhaver

Editor-in-Chief

PLOS Genetics

Comments from the reviewers (if applicable):

**Data Deposition**

http://datadryad.org/submit?journalID=pgenetics&manu=PGENETICS-D-20-00046R2

**Press Queries**

---

## [Editor Report · Acceptance letter]

12 May 2020

PGENETICS-D-20-00046R2 

Glucose transporter 10 modulates adipogenesis via an ascorbic acid-mediated pathway to protect mice against diet-induced metabolic dysregulation 

Dear Dr Lee, 

We are pleased to inform you that your manuscript entitled "Glucose transporter 10 modulates adipogenesis via an ascorbic acid-mediated pathway to protect mice against diet-induced metabolic dysregulation" has been formally accepted for publication in PLOS Genetics! Your manuscript is now with our production department and you will be notified of the publication date in due course.

With kind regards,

Matt Lyles

PLOS Genetics

On behalf of:
